# A plasma membrane template for macropinocytic cups

**Douwe M Veltman[1,2]\*, Thomas D Williams[1], Gareth Bloomfield[1], Bi-Chang Chen[3‡], Eric Betzig[3], Robert H Insall[2†], Robert R Kay[1†]**

[1]MRC Laboratory of Molecular Biology, Cambridge, United Kingdom; [2]Beatson Institute for Cancer Research, Glasgow, United Kingdom; [3]Janelia Farm Research Campus, Howard Hughes Medical Institute, Ashburn, United States

**Abstract** Macropinocytosis is a fundamental mechanism that allows cells to take up extracellular liquid into large vesicles. It critically depends on the formation of a ring of protrusive actin beneath the plasma membrane, which develops into the macropinocytic cup. We show that macropinocytic cups in *Dictyostelium* are organised around coincident intense patches of $PIP_3$, active Ras and active Rac. These signalling patches are invariably associated with a ring of active SCAR/WAVE at their periphery, as are all examined structures based on $PIP_3$ patches, including phagocytic cups and basal waves. Patch formation does not depend on the enclosing F-actin ring, and patches become enlarged when the RasGAP NF1 is mutated, showing that Ras plays an instructive role. New macropinocytic cups predominantly form by splitting from existing ones. We propose that cup-shaped plasma membrane structures form from self-organizing patches of active Ras/$PIP_3$, which recruit a ring of actin nucleators to their periphery.

**\*For correspondence:**
douweveltman@gmail.com

[†]These authors contributed equally to this work

**Present address:** [‡]Research Center for Applied Science, Academia Sinica, Taipei, Taiwan

**Competing interests:** The authors declare that no competing interests exist.

## Introduction

Macropinocytosis provides cells with an efficient way of taking up large volumes of medium into intracellular vesicles, from which they can extract nutrients, antigens and other useful molecules (*Bloomfield and Kay, 2016*; *Egami et al., 2014*; *Maniak, 2001*; *Swanson, 2008*; *Swanson and Watts, 1995*). It is an ancient process, used for feeding by amoebae (*Hacker et al., 1997*; *Thilo and Vogel, 1980*), but one that is important for a wide spectrum of human biology, including uptake of drugs, and large-scale sampling of extracellular medium for antigens by immune cells. It has also been hijacked by pathogens as a major route of entry (*Mercer and Helenius, 2012*). Recent data suggest that macropinocytosis is a principal and widely used method for sustaining the excessive metabolic demands of cancer cells (*Commisso et al., 2013*; *Kamphorst et al., 2015*) and may be implicated in the spread of neurodegenerative disease within the brain (*Münch et al., 2011*).

Considering its biological importance, macropinocytosis is not well understood. Macropinosomes form from cup-shaped extensions of the plasma membrane, often known as circular ruffles, which are extended by actin polymerisation. The leading rims of these ruffles must be driven outwards to enclose liquid – often for a very significant fraction of the cell's diameter – but the base must be held static. The resulting cups can be several microns in diameter, and eventually close by constriction of their rim, with membrane fusion producing an endocytic vesicle. Here we address a critical and mysterious question about this process - how do cells organize actin to polymerize in a ring and so form the walls of the cup?

In the closely related process of phagocytosis, in which solid particles are taken up, it is proposed that cup formation is guided by engaging receptors with the particle to be engulfed in a zippering process (*Freeman and Grinstein, 2014*; *Griffin et al., 1975*). However, macropinosomes take up fluid and so cannot use a particle as a template in this way. Nor is there any known equivalent in

**eLife digest** Cells can use a process known as macropinocytosis to take up fluid from their surroundings. This process plays an important role in many situations. For example, it allows human immune cells to sample their environment to search for harmful microbes and viruses and helps cancer cells to collect more nutrients so that they can grow more rapidly. During macropinocytosis, a protein called actin – which provides structural support to cells – drives the formation of cup-shaped structures from the membrane that surrounds the cell. Several signaling molecules control when and where the "cups" form, but it was not known exactly how the different types of molecules work together.

Here Veltman et al. used a technique called lattice light sheet microscopy to investigate how the macropinocytic cups form in a single-celled amoeba known as *Dictyostelium*. The experiments revealed that to make a cup, the actin first arranges to form a ring. The ring copies a template in the membrane, which consists of high concentrations of signaling molecules, and then extends outward to form a hollow cup by which fluid is taken up. The most important signaling molecule identified in these patches of membrane is a protein called Ras, which is mutated and hyperactive in many different types of cancer. In *Dictyostelium* cells that have a genetic mutation that makes Ras more active, the patches of signaling molecules and macropinocytic cups were larger than in normal cells.

The findings of Veltman et al. provide new details about how cells engulf fluids from their surroundings. The next steps will be to investigate how the signaling molecules form patches in the first place, and how they attract actin molecules. Also, more research is necessary to find out whether all cells take up fluid in a similar way or if other methods have evolved in mammalian cells.

macropinocytosis of the coat that organises clathrin-mediated endocytosis. Thus it appears that macropinocytic cups must form by a self-organizing process within the actin polymerization machinery and its regulators.

The dynamic actin that polymerises around macropinocytic cups is probably initiated by a number of nucleators, including both formins, such as ForG, which is needed for the basal part of phagocytic cups in *Dictyostelium* (*Junemann et al., 2016*), and the Arp2/3 complex (*Insall et al., 2001*), which produces dendritic structures (*Pollard and Borisy, 2003*), like the actin that drives pseudopods. Assembly of Arp2/3 based actin is controlled by the WASP family of nucleation promoting factors; the two family members that act at the plasma membrane are WASP and SCAR/WAVE (hereafter called SCAR). WASP is important for actin polymerisation during clathrin-mediated endocytosis (*Taylor et al., 2011*), and SCAR, acting in a five-membered complex (*Eden et al., 2002*), for the formation of pseudopods (*Seastone et al., 2001*; *Veltman et al., 2012*). It is not known which is responsible for macropinocytosis.

Ras and phosphoinositide signalling help organize the cytoskeleton for macropinocytosis and phagocytosis (*Bar-Sagi and Feramisco, 1986*; *Bloomfield and Kay, 2016*; *Bohdanowicz and Grinstein, 2013*; *Rodriguez-Viciana et al., 1997*; *Swanson, 2014*). There is evidence that Ras activity stimulates macropinocytosis in both mammalian and *Dictyostelium* cells, and macropinocytic cups are associated with an intense domain, or 'patch', of $PIP_3$ (*Araki et al., 2007*; *Parent et al., 1998*; *Yoshida et al., 2009*), which is essential for their function (*Araki et al., 1996*; *Buczynski et al., 1997*; *Hoeller et al., 2013*; *Zhou et al., 1998*).

In macrophages, which often evolve macropinocytic cups from linear ruffles, it has been suggested that ruffle circularisation creates a diffusion barrier in the membrane leading to intensified $PIP_3$ signalling and a domain of $PIP_3$ in the centre of the circular ruffle (*Welliver et al., 2011*). This domain then drives the further progression of the macropinocytic cup.

In axenic strains of *Dictyostelium*, which grow efficiently in liquid medium, macropinocytosis is massively up-regulated due to mutation of the RasGAP neurofibromatosis-1 (NF1) (*Bloomfield et al., 2015*; *Hacker et al., 1997*; *Kayman and Clarke, 1983*). These strains are thus an excellent starting point for research into the organising principles behind macropinocytic cup formation. Here we examine macropinocytosis with unprecedented 3D detail using lattice light sheet microscopy, and map the spatial and temporal control of actin regulators such as SCAR and WASP

with respect to signalling molecules including $PIP_3$ and active Ras. This leads us to propose a new and general hypothesis for the formation of cups from the plasma membrane.

## Results

### The origins of macropinosomes in axenic strains of *Dictyostelium*

To determine whether macropinosomes form in *Dictyostelium* by circularization of linear ruffles, as reported for macrophages (*Welliver and Swanson, 2012*), we used lattice light sheet microscopy (*Chen et al., 2014*), which allows unparalleled high-resolution imaging of light-sensitive and dynamic cells over prolonged periods. In axenic cells expressing an F-actin reporter, three types of large F-actin structure are routinely detected: macropinocytic cups, which predominate, pseudopods and basal waves.

3D movies show that the majority of macropinocytic cups initiate by splitting of existing ones (62%; n = 152, *Figure 1E*). Splitting occurs by a variety of routes, including: simple division in the middle; detachment of a small ruffle that grows into a new macropinocytic cup (*Figure 1A*, *Video 1*); and abortive fragmentation of a parental macropinocytic cup into multiple daughter cups. We examined a reporter for active Rac in some of these movies and found that it is spatiotemporally associated tightly with F-actin in this morphological process (*Figure 1B*, *Video 2*).

The remaining macropinocytic cups form de novo, expanding from places where no previous F-actin activity was detected. In more than 90% of cases the initiation is close to the base of the cell, even though most mature macropinocytic cups are present on the top, and are commonly described as 'crowns'. In the example illustrated in *Figure 1C* (*Video 3*), the parental ruffle first emerges close to the substratum (t = 0, white arrow) and cannot unequivocally be classified as either pseudopod or circular ruffle. The ruffle then quickly sweeps to the top of the cell, during which time it grows in size and splits several times, to produce multiple full-grown macropinocytic cups. As with splitting macropinocytic cups, the F-actin in de novo macropinosome cups is closely associated with signalling molecules, as illustrated by active Ras in *Figure 1D* (*Video 4*) and discussed later. Circular ruffles can persist on the cell surface for prolonged periods before either closing successfully or regressing back into the cell body. Closure of the cup can be quite abrupt and often appears to involve the inward collapse of the rim (*Figure 1F*, *Video 5*) (*Swanson et al., 1999*).

The other large F-actin projections in growing *Dictyostelium* cells are pseudopods. These are distinguished from macropinocytic circular ruffles by their shape, which is convex instead of concave. De novo pseudopods also initiate close to the substratum (*Figure 1G*, white arrow and *Video 6*) and expand steadily to their full size. Pseudopods are surprisingly rare in growing axenic cells, accounting for less than 5% of all large F-actin structures.

Finally, we could follow the enigmatic actin waves that move across the basal surface of vegetative cells (*Bretschneider et al., 2004*, *2009*; *Gerisch, 2010*). These waves also generally originate from existing ruffles by splitting. When the splitting ruffle in *Figure 1H* (*Video 7*) contacts the substratum, it initiates an actin wave that spreads across the entire footprint of the cell, becoming so dominant that other large F-actin structures are suppressed and flattening the cell into a smooth bell-shape.

These observations show that the large macropinocytic cups of axenic *Dictyostelium* cells generally form by splitting or by expanding de novo from a small focus, as in fibroblasts (*Bernitt et al., 2015*), rather than by circularization of linear ruffles (*Welliver and Swanson, 2012*). The smaller macropinocytic cups of wild-type cells (see later) also more normally form de novo or by splitting, rather than by circularization.

### $PIP_3$, Ras and SCAR are required for normal fluid phase uptake

We tested the involvement of PIP3 and Ras signalling in macropinocytosis using an isogenic set of mutants in which we measured both fluid uptake and growth in liquid medium (Supplementary material, *Table 1*). Either increased or decreased $PIP_3$ levels (PTEN and PI3-kinase mutants) are deleterious to fluid uptake and growth in liquid medium, as expected from earlier work (*Clark et al., 2014*; *Hoeller and Kay, 2007*). Two independent RasG- mutants are substantially impaired in growth in liquid medium, as previously described, but contrary to the earlier report (*Khosla et al., 2000*), both are also defective in fluid uptake. Compensation by other Ras proteins and genetic background

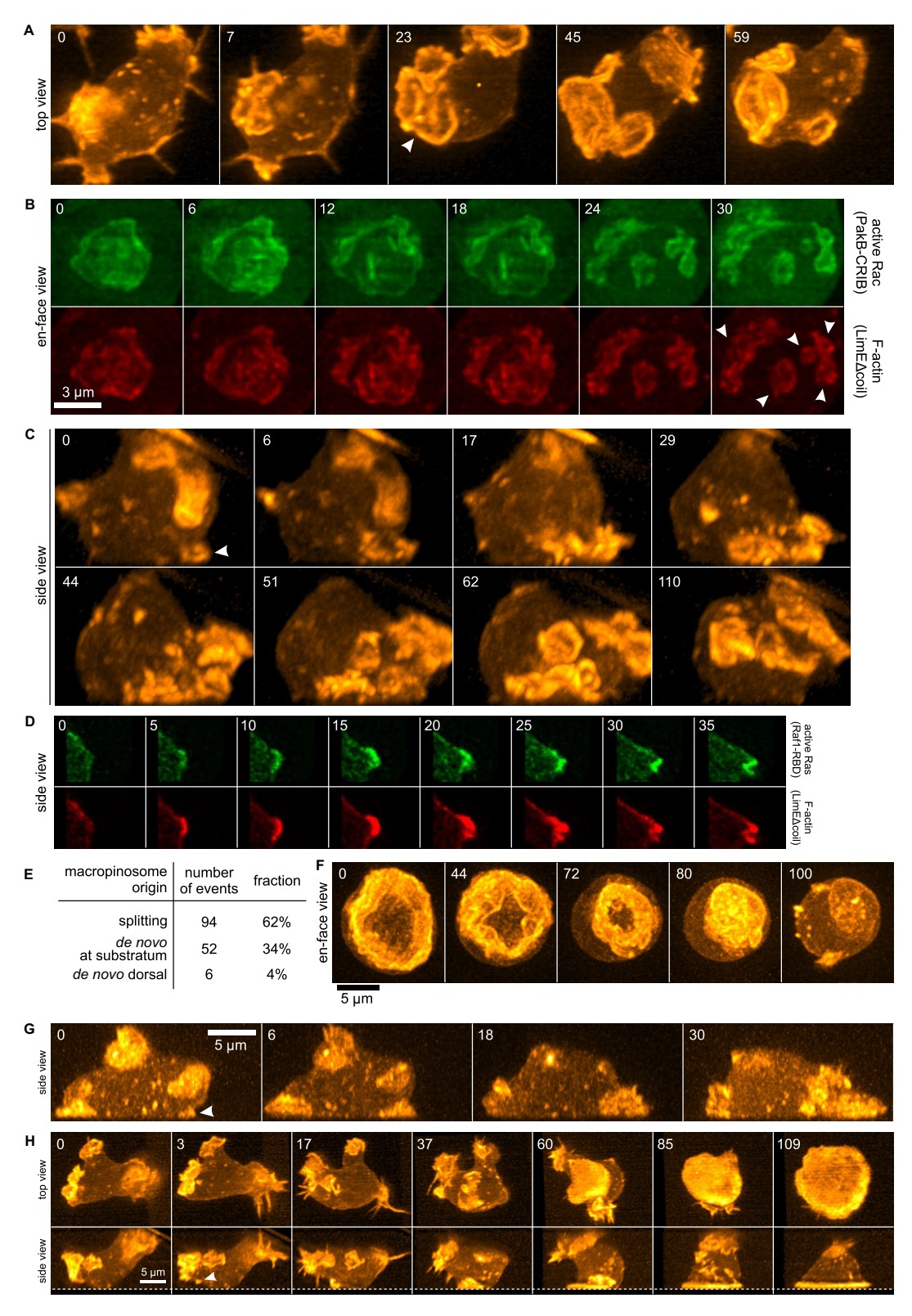

**Figure 1.** Life histories of macropinocytosis, pseudopods and basal waves. Vegetative axenic cells of strain Ax2 were followed in 3D time-lapse movies made by lattice light sheet microscopy. Images show maximum intensity projections. Cells express a marker for F-actin (LimEΔcoil-RFP) unless otherwise indicated. Numbers indicate time in seconds. (A) A new macropinocytic cup formed by splitting (arrow marks nascent daughter macropinocytic cup). (B) Close-up of a large macropinocytic cup, viewed *en-face*, that fragments and forms multiple smaller macropinocytic cups (arrowed). (C) Origin of a

*Figure 1 continued on next page*

*Figure 1 continued*
cluster of macropinocytic cups from a small basal F-actin structure (arrow). (**D**) Detail of a cell initiating a de novo dorsal macropinocytic cup (ie not in contact with substratum). (**E**) Table of macropinocytic cup origins. (**F**) Close-up of macropinocytic cup closure, viewed *en face*. (**G**) Growth of a pseudopod from a small F-actin structure close to the substratum (arrow). (**H**) Growth of a basal F-actin wave from a small F-actin punctum (arrow), to eventually encompass the entire basal surface of the cell. See also supplementary movie 1-7 for full movies of panel **A**, **B**, **C**, **D**, **F**, **G** and **H** respectively.

differences may account for the discrepancy (*Bloomfield et al., 2008*; *Bolourani et al., 2010*). RasC null cells have no growth defect and a lesser defect in fluid uptake, while RasS (not tested here) may also contribute to macropinocytosis (*Chubb et al., 2000*). Notably, we confirm that the Arp2/3 activator, SCAR, is required for efficient fluid uptake (*Seastone et al., 2001*).

## A ring of active SCAR forms around PIP$_3$ domains at the rim of macropinocytic cups

The SCAR complex is mostly cytosolic and basally inactive, but when recruited to the plasma membrane it causes actin polymerization through the Arp2/3 complex (*Steffen et al., 2004*; *Ura et al., 2012*). A GFP reporter tagged at the HSPC300 subunit accumulates at sites of actin polymerization (*Veltman et al., 2012*). To confirm that this accumulation signifies the presence of activated SCAR complex, we correlated the signal with the expansion of pseudopods, using this as a proxy for actin polymerization. The results clearly show that the reporter is recruited during expansion phases but lost in stalls (*Figure 2A,B*). This correlation holds true globally: SCAR reporter intensity along the membrane correlates well with the local membrane expansion speed (*Figure 2C*) as all pixels with high SCAR reporter are associated with positive instantaneous membrane speed. Note that the small set of pixels with very high membrane speeds and no SCAR (*Figure 2C*, red arrow) are due to blebs (*Zatulovskiy et al., 2014*).

In images recorded in 3D, the reporter reveals a thin, sometimes broken ring of active SCAR around the lip of macropinocytic cups (*Figure 2D* and *Video 8*). The presence of a ring could not be predicted by imaging the actin cytoskeleton itself, as actin filaments are distributed rather uniformly throughout the cup (*Figure 2H*). These circular SCAR structures are not seen in pseudopods, where 3D images show the same discrete blocks of SCAR as in 2D images (*Figure 2E*).

The discovery of these remarkable rings immediately raises the question of how individual SCAR molecules are coordinated to maintain the ring shape. We visualised PIP$_3$ using a double reporter that co-expresses the PH-domain of CRAC fused to RFP (*Insall et al., 1994*). This revealed a

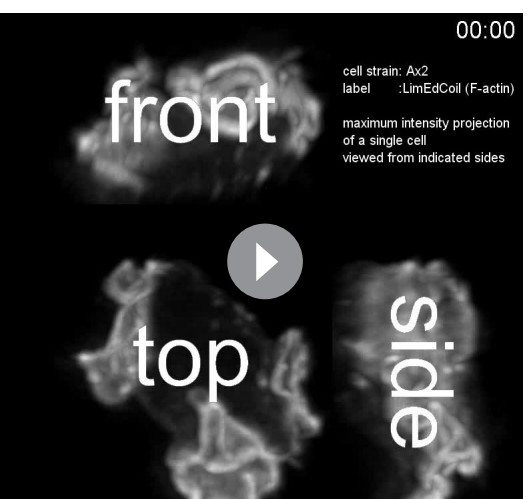

**Video 1.** Macropinosomes are generated by splitting. A vegetative Ax2 cell expressing an F-actin marker imaged using lattice light sheet microscopy and viewed from three perpendicular angles.

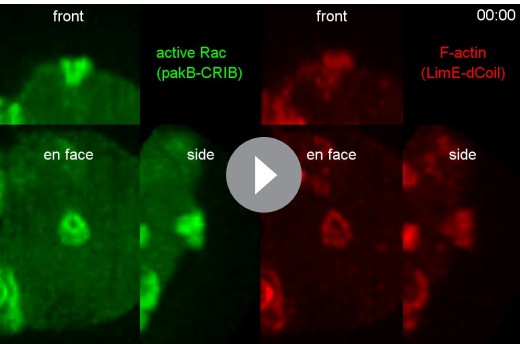

**Video 2.** Macropinocytic cups are generated by splitting. Detail of a vegetative Ax2 cell expressing markers for active Rac and F-actin, imaged using lattice light sheet microscopy and viewed from three perpendicular angles.

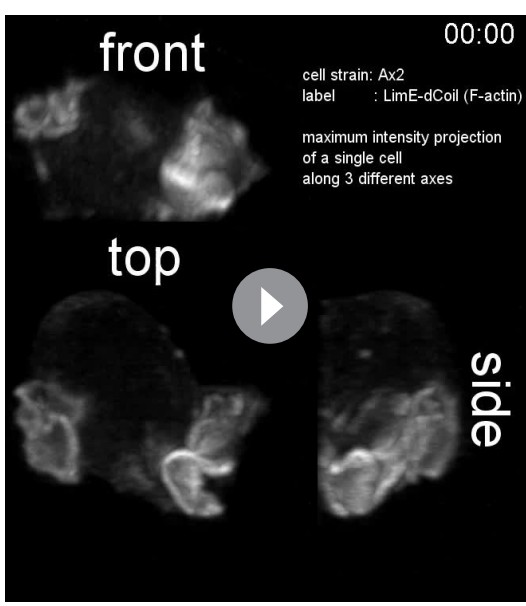

**Video 3.** Macropinocytic ruffles often initiate on the substratum. A vegetative Ax2 cell expressing an F-actin marker imaged using lattice light sheet microscopy and viewed from three perpendicular angles.

second remarkable feature of SCAR rings: they follow the edges of intensely stained domains, or 'patches' (*Postma et al., 2004*) of $PIP_3$. In all macropinocytic cups examined, of whatever size, the concave cup contains a patch of $PIP_3$ and SCAR is present as a ring around this patch, without detectable recruitment to its centre (*Figure 2F and G* and *Video 9*).

This was confirmed in a larger sample by measuring fluorescence intensity at the centre and rim of 17 macropinocytic cups from nine cells rendered in 3D (*Figure 3A–B*). All cup centres contained high levels of $PIP_3$, but SCAR consistently followed the rim of the cup with the mean fluorescence of the SCAR reporter significantly higher than cytosolic background ($p<0.01$), while signal at the centre of the $PIP_3$ patch was not statistically different from the cytosolic background. This is also clear in 2D images, but is easily overlooked as the narrow SCAR ring appears only as tiny puncta in the cross sections obtained from confocal microscopy.

We further tested the spatial relation between $PIP_3$ and SCAR in two ways not requiring visual recognition of macropinocytic cups. In the first, a number of growing axenic cells was analysed as follows. Membrane areas with fluorescence intensity of the $PIP_3$ reporter greater than cytosolic background plus one standard deviation were defined as $PIP_3$ patches, and the associated SCAR signal was measured. In all cases SCAR is consistently and significantly enriched at patch edges ($p<0.01$), and never at their centres (*Figure 3C–D*). In the second test, the fluorescence intensity of the SCAR and $PIP_3$ reporters was extracted from the circumference of a number of growing cells and the results plotted as a 2D histogram (*Figure 3E*). In pixels with high $PIP_3$ signal, the SCAR signal is low, and conversely in pixels with high SCAR, $PIP_3$ is low. This method cannot show whether high SCAR and $PIP_3$ pixels are adjacent, but it does confirm that SCAR and $PIP_3$ do not co-localise but instead are anti-correlated.

## SCAR is associated to the periphery of $PIP_3$ patches throughout macropinocytic cup lifetime

The complete lifetime of a de novo macropinocytic cup is shown in *Figure 4A* (*Video 10* shows another example). The $PIP_3$ patch first becomes visible at $t = 1$ and this sub-micron sized patch is already flanked by puncta of SCAR. As the patch of $PIP_3$ grows the SCAR puncta remain dynamically associated with its edge right up to closure of the macropinocytic cup, after which the SCAR signal quickly disappears and the vesicle is internalised. The SCAR is not detected at the centres of the patches above background.

This is also shown in a kymograph of the membrane pixels of a single cell as it makes several macropinosomes (*Figure 4B*). The SCAR signal, though sometimes weak, can be traced along the edge of the $PIP_3$ patches from the start of a patch to its abrupt loss at invagination. Combining the data from several macropinocytosis

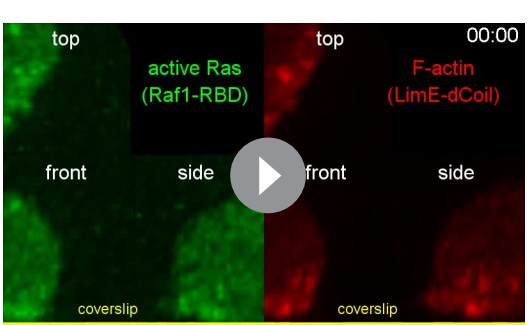

**Video 4.** A small fraction of de novo macropinocytic cups are initiated off- substratum. Detail of a vegetative Ax2 cell expressing markers for active Ras and F-actin, imaged using lattice light sheet microscopy and viewed from three perpendicular angles.

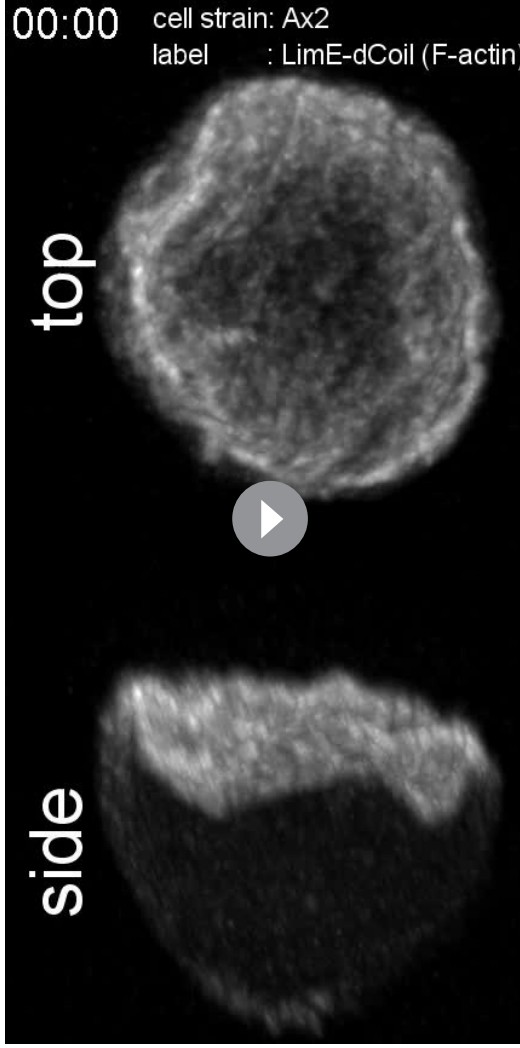

**Video 5.** Closure of a macropinocytic cup. A vegetative Ax2 cell expressing an F-actin marker imaged using lattice light sheet microscopy and viewed from two perpendicular angles.

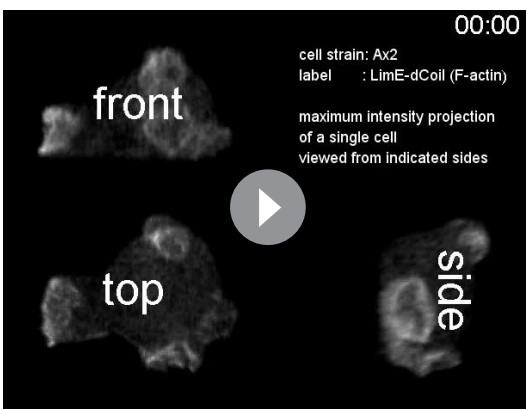

**Video 6.** Pseudopods are distinct from macropinocytic ruffles. A vegetative Ax2 cell expressing an F-actin marker imaged using lattice light sheet microscopy and viewed from three perpendicular angles. A pseudopod is initiated on the right hand side of the cell at t = 1 min.

events confirms this continuous association (*Figure 4C–E*). Thus despite changes in size and shape of PIP$_3$ patches, SCAR remains associated with their edges, and only their edges, throughout the macropinocytic cup lifetime.

## All PIP$_3$ patches, whatever their origin, recruit SCAR to their periphery

It seemed possible that as a rule of cytoskeletal organization in *Dictyostelium*, PIP$_3$ patches always recruit SCAR to their edges. Four other examples of PIP$_3$ patches support this: two from growing cells, and two from starved cells, which are highly migratory, chemotactically sensitive, and morphologically very different from growing cells:

During phagocytosis *Dictyostelium* cells make a PIP$_3$ patch where they contact the particle to be ingested (*Clarke et al., 2010*; *Marshall et al., 2001*). In the yeast case shown in *Figure 5A*, SCAR is recruited to the edges of the PIP$_3$ patch, and not the centre, while a 3D view reveals a full ring of SCAR around the rim of the phagocytic cup. Indeed, a clear ring of SCAR could be detected in all such cases, provided expression of the SCAR reporter was low enough to avoid excessive background fluorescence.

Basal waves have a core of PIP$_3$ surrounded by a ring of F-actin (*Bretschneider et al., 2004*, *2009*; *Gerhardt et al., 2014*; *Gerisch, 2010*), and again, the basal PIP$_3$ patches are invariably surrounded by a ring of SCAR (*Figure 5B*). Basal waves are favourable for microscopy, and we found that WASP is also excluded from PIP$_3$ patches and forms a ring around them, though weaker and less coherently than SCAR (*Figure 5C*). Similarly, WASP forms rings at the edges of PIP$_3$ patches of normal macropinocytic cups, again more weakly than SCAR (*Figure 5D*). The remaining Arp2/3 activator in *Dictyostelium*, WASH, does not associate with PIP$_3$ patches (*Figure 5—figure supplement 1*).

**Table 1.** Growth and fluid uptake by mutants.

| Strain | Mutated protein | Genotype | MGT±SEM (hr) | Fluid uptake ± SEM (nl/10six cells/h) |
|---|---|---|---|---|
| Ax2 | | parental | 9.26 ± 0.26 (26) | 114.7 ± 10.8 (9) |
| HM1505 | RasC | *rasC-* | 9.27 ± 0.26 (7) | 84.3 ± 6.3 (3) |
| HM1497 | RasG | *rasG-* | 27.6 ± 2.9 (7) | 55.3 ± 8.4 (3) |
| HM1514 | RasG | *rasG-* | 19.1 ± 0.24 (3) | 67.6 ± 1.9 (3) |
| HM1200 | PI3K1-5 | *pikA-, pikB-, pikC-, pikF-, pikG-* | 103 ± 1.1 (3)* | 9.0 ± 0.6 (3) |
| HM1289 | PTEN | *ptenA-* | 19.0 ± 1.4 (9) | 12.4 ± 0.6 (3) |
| HM1809 | SCAR | *scrA-* | 27.8 ± 2.4 (6) | 56.8 ± 9.7 (3) |
| HM1818 | SCAR | *scrA-* | 38.4 ± 4.6 (10) | 19.7 ± 5.0 (3) |

* taken from *Hoeller et al. (2013)*.

During chemotactic aggregation, developing cells form small chains and streams with strong head-to-tail adhesions between them and PIP$_3$ patches in their front (*Dormann et al., 2002*). These patches are invariably surrounded by a ring of SCAR. In the example shown in *Figure 5E–G*, a cell strongly expressing reporters is situated between two poorly expressing cells. The strongly expressing cell forms a PIP$_3$ contact patch, with SCAR present as a clear ring and excluded from the centre.

Cells respond to cyclic-AMP by making PIP$_3$, initially homogenously and then, after about a minute, in patches at the membrane (*Postma et al., 2004*). In the low light conditions required for time-lapse imaging, the SCAR signal is weak, but where detected, it is clearly at the edges of the PIP$_3$ patches (*Figure 5H–J* and *Video 11*). These patches have sometimes been regarded as new pseudopods (for example [*Chen et al., 2003*]), but many become concave and close to engulf a drop of medium, indicating that the cell is performing macropinocytosis, not a chemotactic response.

## PIP$_3$ patches are based on active Ras but do not require F-actin ruffles

PIP$_3$ is largely made by Ras-activated PI3-kinases (*Clark et al., 2014*; *Funamoto et al., 2002*; *Hoeller and Kay, 2007*). We confirmed that a patch of activated Ras exactly coincides with each PIP$_3$ patch (*Figure 6A*) (*Sasaki et al., 2004*; *2007*). Similarly, plots of intensity, pixel-by-pixel, show exceptional correlation between the Ras and PIP$_3$ signals (*Figure 6B*). Thus PIP$_3$ patches have a matching patch of activated Ras, which could sustain them by activating PI3-kinase.

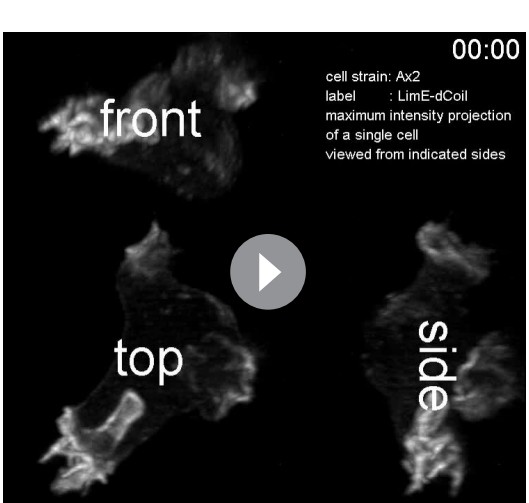

**Video 7.** Basal F-actin wave that originates by splitting from a nascent macropinosome. A vegetative Ax2 cell expressing an F-actin marker imaged using lattice light sheet microscopy and viewed from three perpendicular angles. Image jitter in this movie was due to technical issues with the microscope's Z-drive).

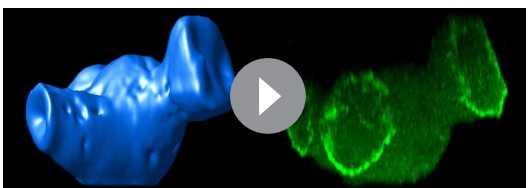

**Video 8.** The rim of macropinocytic cups is traced by a thin line of SCAR. Shown is a vegetative Ax2 cell expressing the SCAR marker HSPC300. A Z-stack was collected using a spinning disk microscope and deconvolved using a calculated point spread function. Left panel shows a surface render of the cell outline and the right panel shows a 3D reconstruction of the fluorescence signal using maximum intensity projection.

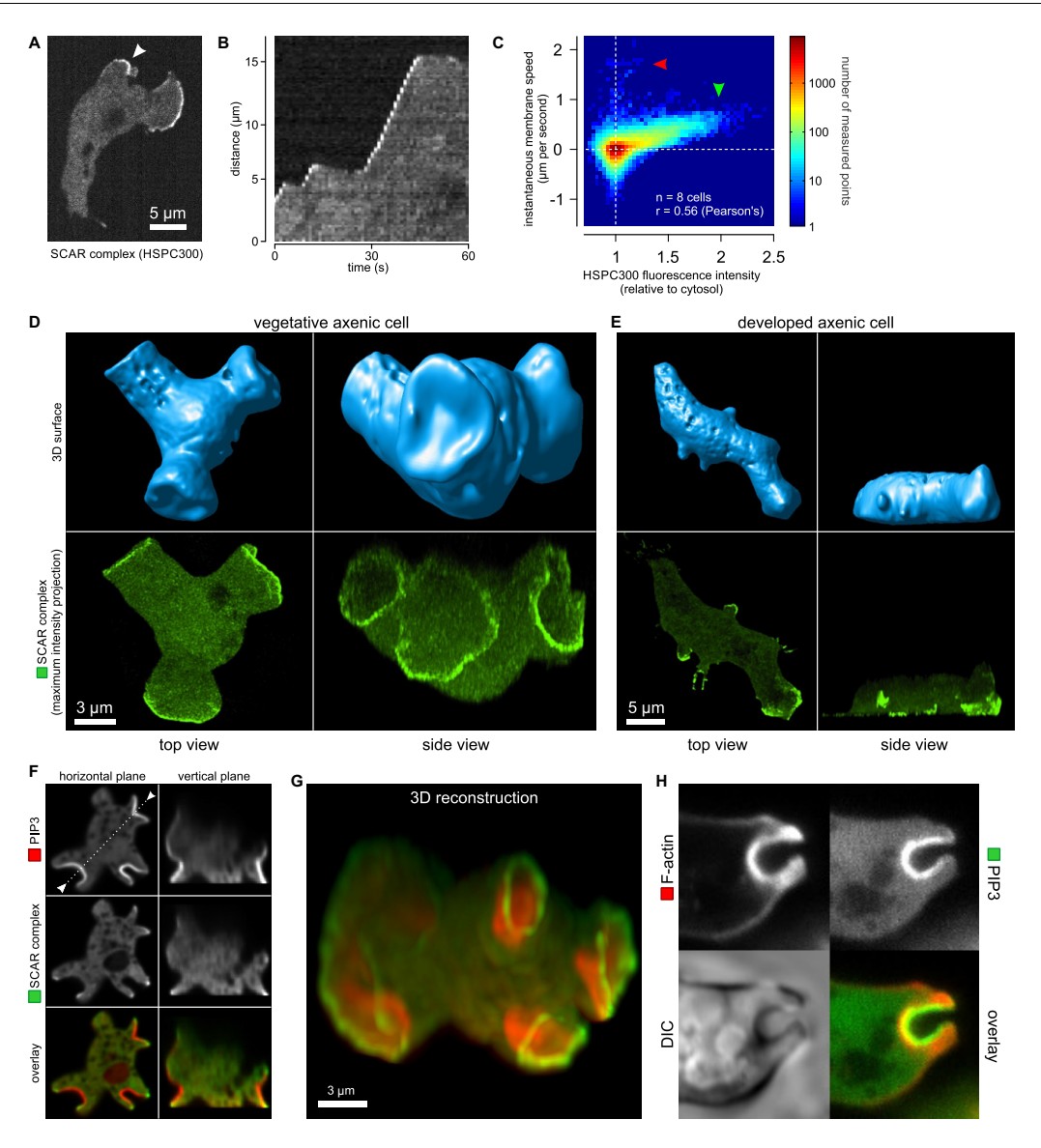

**Figure 2.** Macropinocytic cups contain a central domain of PIP$_3$ surrounded by a ring of SCAR. (**A**, **B**, **C**) Evidence that fluorescently tagged SCAR complex faithfully marks regions of active actin polymerisation: (**A**) An aggregation-competent cell moving under an agarose overlay (optimal conditions for visualising pseudopods) showing HSPC300-GFP recruited to the two pseudopods; (**B**) Kymograph of expansion of the pseudopod arrowed in (**A**) showing that the SCAR reporter is present during periods of expansion, but absent in the plateaus when the pseudopod is not expanding; (**C**) Membrane speed and SCAR complex accumulation are positively correlated. The HSPC300-GFP signal and local membrane speed was measured at 100 points along the membrane of a motile cell. Data of eight independent cells was combined and plotted as a 2D histogram. Green arrow indicates data points with high levels of SCAR and positive displacement. Red arrow indicates data points due to blebs, which are actin-free and expand much faster than pseudopods (*Zatulovskiy et al., 2014*). (**D**) SCAR is recruited as a ring to the lip of macropinocytic cups. The upper panels show top and side views of a surface rendering of a cell with three macropinocytic cups and the lower shows the same cell with a SCAR reporter. (**E**) SCAR is recruited to pseudopods in distinct blocks, not as a ring. Pitted appearance of the 3D surface is a rendering artefact caused by small vesicles that reside just underneath the cell membrane. (**F**) SCAR is recruited to the edge of an intense PIP$_3$ patch in the macropinocytic cup. The white dotted line in the left panel corresponds to the position of the vertical plane in the right panel. (**G**) 3D reconstruction of the cell in the previous panel. (**H**) F-actin is nearly uniformly distributed in the macropinocytic cup and does not predict the localization of SCAR. Ax2 cells were used in all panels. HSPC300 was used as a marker for the SCAR complex, PH-CRAC as a reporter for PI(3,4,5)P$_3$ and Lifeact as a reporter for F-actin. 3D images were reconstructed from Z-stacks taken on a spinning disk microscope.

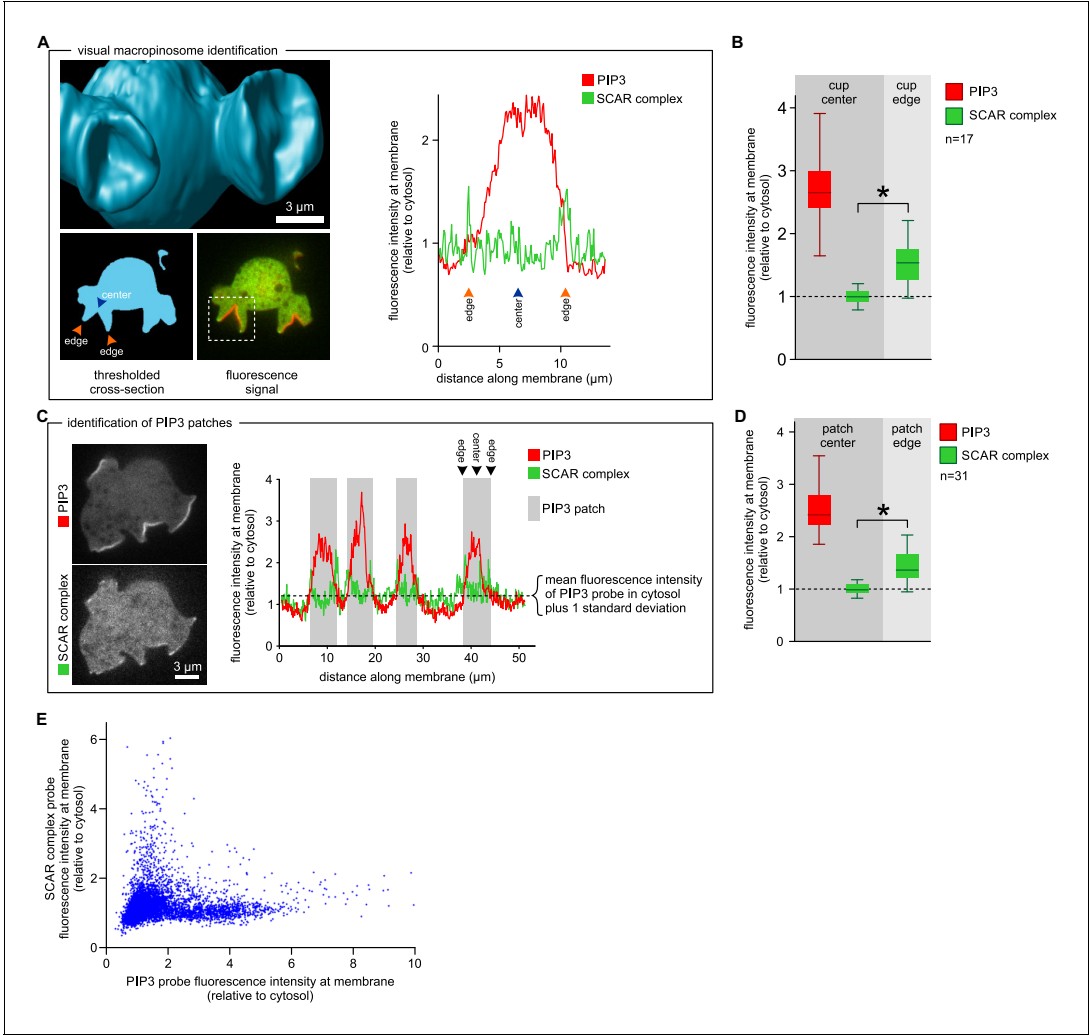

**Figure 3.** SCAR is present at the edge but not the centre of the macropinocytic cup. (**A**, **B**) Quantification of $PIP_3$ and SCAR in macropinocytic cups identified visually. (**A**) Method of analysis: Macropinocytic cups were identified morphologically from 3D rendered images and the position of their center (blue arrow) and edge (orange arrows) was noted. Fluorescence intensity along the cell membrane of the boxed macropinocytic cup was plotted and the intensity was measured at the marked center and edge. (**B**) $PIP_3$ and SCAR fluorescence intensity at center and edge of 17 visually identified macropinocytic cups. Bar indicates the mean, box indicates the second and third quartile. Whiskers indicate the range of the data. (**C**, **D**) Analysis of SCAR in $PIP_3$ patches. (**C**) Method of $PIP_3$ patch analysis: Membrane-bound fluorescence intensity of the respective markers was measured for individual cells. $PIP_3$ patches were defined as those membrane regions where the fluorescence intensity is greater than mean cytosol plus one standard deviation (dotted line) and these regions are marked in grey. (**D**) Quantification of fluorescence intensity at the centers and edges of 31 identified $PIP_3$ patches. Bar indicates the mean, box indicates the second and third quartile. Whiskers indicate the range of the data. (**E**) Anti-correlation between SCAR and $PIP_3$ reporter intensity. In this analysis the intensity of fluorescence in all membrane pixels was compared, irrespective of the morphological structure in which they lay. The plot shows the combined data from 16 cells. Vegetative Ax2 cells expressing the SCAR complex reporter HSPC300-GFP and the PI(3,4,5)$P_3$ reporter PH-CRAC-mRFP were used in all experiments. The asterisk marks significant differences (p<0.01).

Similarly, the Ras/$PIP_3$ patch overlaps a patch of active Rac1, as detected by the CRIB domain (*Figure 6C*) (*Manser et al., 1994*). Rac1 is an upstream regulator of SCAR, and has been implicated in macropinocytosis (*Dumontier et al., 2000*; *Palmieri et al., 2000*). However, its broad distribution cannot simply account for the much narrower SCAR ring. Alternatively, Rac1 may define a permissive area where SCAR can be activated or other Rac isoforms may be involved, such as RacB, RacC or RacG (*Lee et al., 2003*; *Seastone et al., 1998*). No specific markers exist for their activated state, but the RacG molecule itself is modestly enriched at the rim of phagocytic cups (*Somesh et al., 2006*).

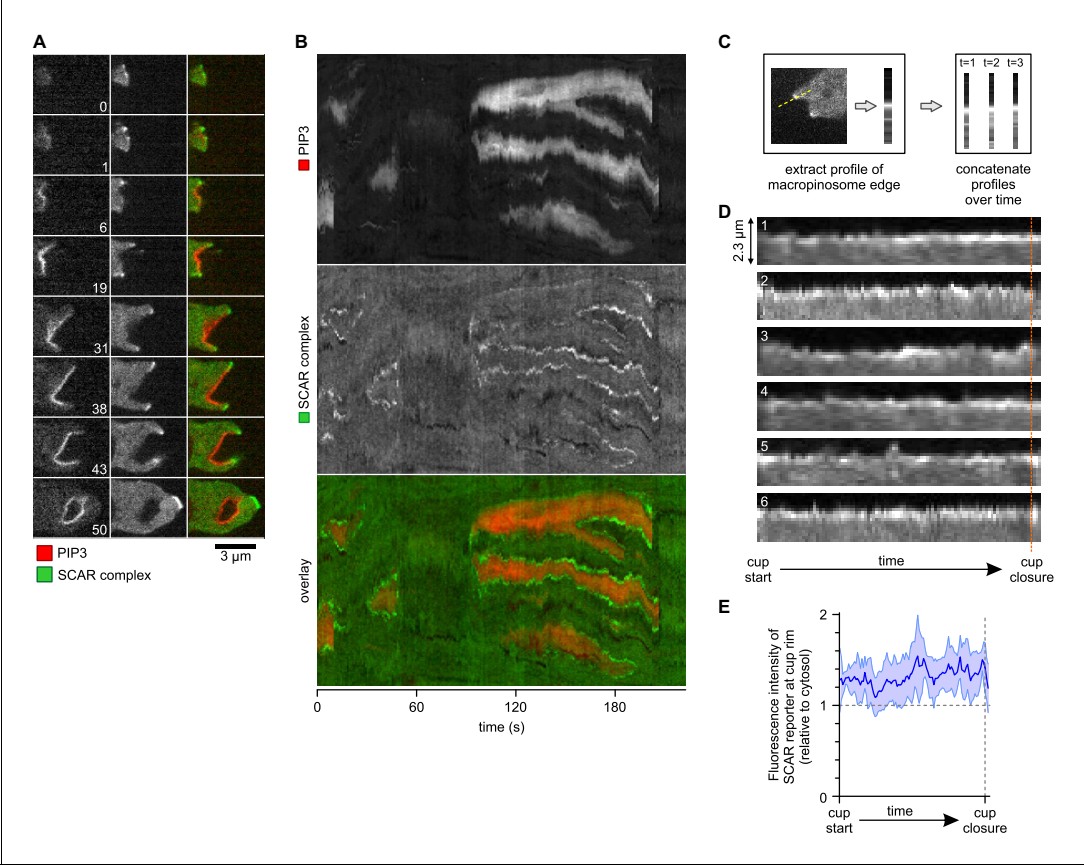

**Figure 4.** A SCAR ring encircles PIP$_3$ patches throughout their lifetime. (**A**) Detail of a cell initiating a de novo macropinocytic cup. Puncta of SCAR flank the PIP$_3$ patch at all times. Time is indicated in seconds. (**B**) Kymograph of the fluorescence intensity along the membrane of a vegetative cell completing several macropinocytosis events. SCAR is associated with the edge of PIP$_3$ patches during their entire lifetime and never with their centers. (**C**) Quantification of peripheral SCAR signal during macropinocytosis. A line was drawn through the edge of the macropinocytic cup and extending into the cytosol for each frame of a time lapse of the lifetime of a complete macropinocytosis event. Line plots were concatenated resulting in kymographs as shown in panel (**D**). Orange dotted line indicates the closure event. Numbers indicate independent macropinocytosis events. (**E**) The SCAR signal at the edge of macropinocytic cups is present from start to finish. The fluorescence intensity at the macropinocytic cup edge was quantified and averaged for all six analysed macropinocytic cups. Vegetative Ax2 cells expressing the SCAR complex reporter HSPC300-GFP and the PI(3,4,5)P$_3$ reporter PH-CRAC-mRFP were used in all experiments. Images are representative of typical macropinocytosis events.

It has been proposed that PIP$_3$ patch formation requires a positive feedback loop where PIP$_3$ activates Ras (*Sasaki et al., 2007*). We tested this by genetically manipulating PIP$_3$ levels (*Clark et al., 2014*; *Hoeller and Kay, 2007*). A mutant without Ras-activated PI3-kinases and producing only 10% of wild-type PIP$_3$ levels still forms patches of activated Ras at a similar frequency to parental cells (*Figure 6D*). The SCAR signal in confocal cross sections of macropinocytic cups is too small for an accurate comparison of SCAR ring formation between mutants and therefore we used basal waves as a proxy for macropinocytic cups. Ras patches on the basal surface of PI3-kinase null cells still exclude SCAR from their centre and recruit a peripheral ring of SCAR as normal, albeit more weakly than in parental cells (*Figure 6E–F*). Conversely, when PIP$_3$ levels are increased 10-fold by eliminating the PTEN phosphatase, the activated Ras domains do not expand correspondingly (*Figure 6G*) and remain associated with rings of SCAR (*Figure 6H*). Thus Ras, rather than PIP$_3$, is the primary determinant of patches and SCAR rings.

It has also been proposed that PIP$_3$ patch formation requires an enclosing circular ruffle to act as a diffusion trap (*Welliver and Swanson, 2012*). We tested this by controlled use of the actin inhibitor latrunculin-A to inhibit ruffle formation. Latrunculin-A at 1 µM leaves some actin polymerisation intact, and at 5 µM abolishes all visible actin filaments, resulting in spherical cells (*Figure 7A*). Neither treatment abolishes the patches of PIP$_3$, which become larger but less numerous, with a

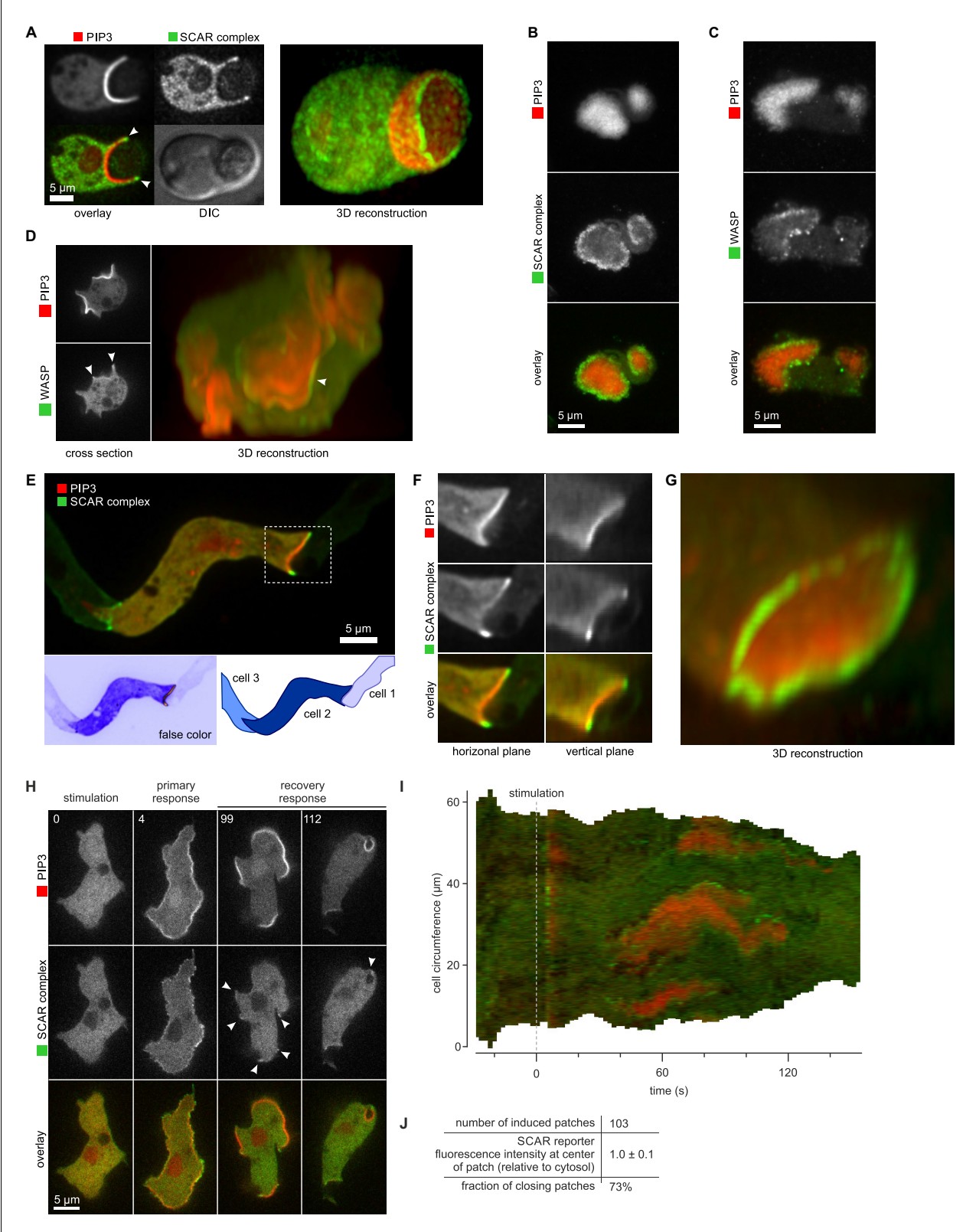

**Figure 5.** SCAR is recruited to the periphery of PIP₃ patches, however they are formed. (**A**) Phagocytosis of a yeast cell. Arrows point to the narrow accumulation of SCAR at the rim of the phagocytic cup. A full 3D render of the same cell is displayed on the right. The yeast cell is not labelled and thus not visible in this image. (**B**) SCAR rings around basal actin waves visualised by TIRF microscopy. These waves consist of a patch of PIP₃ surrounded by a zone of actin polymerisation. PIP₃ patches are invariably surrounded by a ring of SCAR. (**C**) WASP is excluded from the PIP₃ patches of basal waves

*Figure 5 continued on next page*

*Figure 5 continued*

and present as a ring around the patch, but less strongly than SCAR. (D) WASP is present at the edge of PIP$_3$ patches in macropinocytic cups, albeit more faintly than SCAR. Confocal cross sections are shown on the left panels and a full 3D render of the same cell is shown on the right. White arrow indicates the faint rim of WASP around the PIP$_3$ patch. (E–G) SCAR rings at cell-cell contacts. During chemotactic aggregation, streaming cells make head-to-tail attachments and form a domain of PIP$_3$ in their anterior attachment, which is surrounded by a strong SCAR ring; (F) A close-up of the region indicated in the white box in (E); (G) A full 3D reconstruction of the same contact site. (H–J) Response to acute stimulation with the chemoattractant cyclic-AMP. Aggregation-competent cells (equivalent to about 5 hr of starvation) were uniformly stimulated with a saturating dose of cyclic-AMP (1 µM). They respond with a fast and spatially fairly uniform production of PIP$_3$ at 4 s and later with a secondary response in which clear PIP$_3$ patches are formed (panel H, t = 99s). SCAR is uniquely present at the edges of these patches (arrowed) as is most evident in the kymograph (I) and not at the centre (J).

The following figure supplement is available for figure 5:

**Figure supplement 1.** WASH is not recruited to PIP$_3$ patches.

fluorescence intensity not significantly different from control cells (*Figure 7B–E*). We tested whether the sharp boundaries of patches are affected by latrunculin-A by measuring the intensity across the edges of more than 30 patches for each condition, (*Figure 7F–H*). It is clear that latrunculin-A has little effect on the sharpness of the patch, suggesting that a diffusion barrier is not required to maintain its strong spatial coherence.

In summary, a circular ruffle is not essential to create signalling patches, which appear to largely depend on Ras, with PIP$_3$ playing a secondary though still important role.

## The intensity of Ras signalling controls patch and macropinocytic cup size

To test whether Ras plays an instructive role in macropinocytic cup morphogenesis, directly regulating their formation and size rather than acting as a remote trigger or passive participant, we examined the effect of genetically increasing Ras activity. The RasGAP NF1, encoded by the *Dictyostelium axeB* gene, is present in the wild-isolate NC4 but inactivated in its axenic derivatives, including the standard Ax2 used here. We found that macropinocytic cups in NC4 maintain exactly the same organization as in Ax2, with a central patch of PIP$_3$ surround by a ring of SCAR, but are much smaller and shorter-lived and often arise de novo (*Figure 7I,J* and *Videos 12*, *13* and *14*). To confirm that macropinocytic cup size is controlled

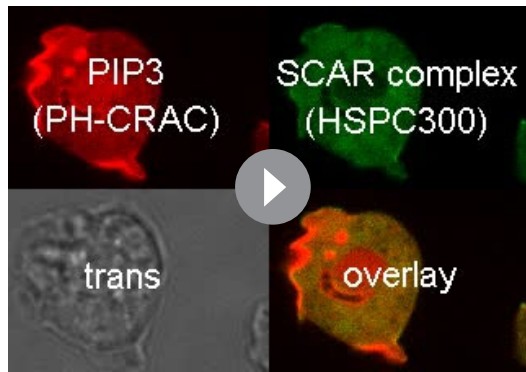

**Video 9.** Macropinocytic cups are defined by a patch of PIP$_3$ that is circumscribed by a thin line of SCAR. Shown is a vegetative Ax2 cell expressing the SCAR marker HSPC300-GFP and the PIP$_3$ marker PH-CRAC-RFP. A Z-stack was collected using a spinning disk microscope and deconvolved using a calculated point spread function. Image shown is a 3D reconstruction of the fluorescence signal using maximum intensity projection.

**Video 10.** SCAR remains dynamically associated to the edge of PIP$_3$ patches throughout macropinocytosis. Vegetative Ax2 cell expressing the SCAR marker HSPC300-GFP and the PIP$_3$ marker PH-CRAC-RFP. Images were taken on a spinning disk microscope.

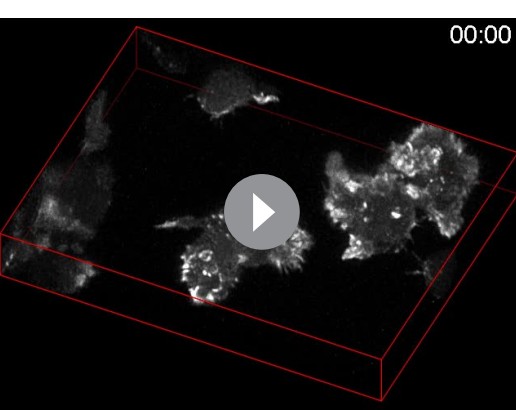

**Video 11.** Chemoattractant-stimulated PIP$_3$ patches recruit SCAR to their edges, not to their centres. A developed Ax2 cell expressing the PIP$_3$ marker PH-CRAC-RFP and the SCAR marker HSPC300-GFP was stimulated at t = 0 with 1 μm cyclic AMP. Images were collected using a spinning disk microscope.

**Video 12.** Large circular ruffles are absent in vegetative wild-type NC4 cells. Shown is a maximum intensity projection of the fluorescence intensity of the F-actin marker LimEΔcoil. Images were taken on a lattice light sheet microscope.

by NF1 we compared an isogenic NF1 knock-out with its parent (DdB; also derived from NC4; [*Bloomfield et al., 2008*]). Cells from each strain were cultivated for 48 hr in axenic medium to maximally induce the rate of macropinocytosis. Under these conditions the *axeB*-null cells that have lost NF1 make significantly larger macropinocytic patches compared to cells from the parental strain (p<0.01, *Figure 7K–L*).

The effect of the loss of NF1 on basal PIP$_3$ patches (basal actin waves) is equally striking. Basal PIP$_3$ patches are prevalent in axenic laboratory strains, especially during early starvation, but absent from all wild-type strains tested (*Figure 7—figure supplement 1* and compare *Videos 15* and *16*). *axeB* knockout cells that have lost NF1 form abundant basal PIP$_3$ patches, but their wild-type parent does not (*Figure 7—figure supplement 1D*). Thus the intensity of Ras signalling governs the size and frequency of SCAR rings in macropinocytic cups and basal waves, showing that Ras must play an instructive role.

We therefore propose that Ras patches, assisted by PI3-kinase and Rac, cause macropinocytic cup formation by recruiting rings of SCAR/WAVE complex to their edge.

## Discussion

Macropinosomes develop from cup-shaped projections of the plasma membrane, whose walls are driven outwards by actin polymerization. They contain a central patch of activated Ras and PIP$_3$ throughout their life and we find that in *Dictyostelium*, this patch is invariably associated with a ring of active SCAR at its edge. We propose that this ring of active SCAR is recruited by the signalling patch and drives a hollow ring of F-actin to extend the walls of the macropinocytic cup.

A possible alternative mechanism comes from immune cells, which make abundant linear ruffles. These occasionally fold back to form circular ruffles, which have been described as diffusion traps that can intensify signalling within them, leading to the formation of a patch of active Ras and PIP$_3$, (*Welliver et al., 2011*). In this model, PIP$_3$ patches form as a consequence of circular ruffle formation, rather than as a cause of it. Despite the evidence that sharply curved membrane areas such as those present at the leading edge of lamellipods can act as a diffusion barrier (*Weisswange et al., 2005*), this idea does not easily extend to *Dictyostelium*, where linear ruffles are much less common, and the central PIP$_3$ patch of macropinocytic cups can still form when ruffle formation is inhibited. However it remains possible that a diffusion barrier forms by a ruffle-independent mechanism, for example by septin-like molecules (*Golebiewska et al., 2011*), or perhaps by cross-linking components within the patch. Further, *Dictyostelium* patches become larger when Ras signalling is increased by NF1 inactivation, showing that Ras plays an instructive part in their formation.

PIP$_3$ patches are coincident with patches of activated Ras, which presumably support them by activating PI3-kinase, and also of activated Rac. Previous work suggest that patches are self-organising structures, which can form independently of input from G-protein coupled receptors (*Sasaki et al., 2007*) and are likely dependent on positive feedback loops between their components

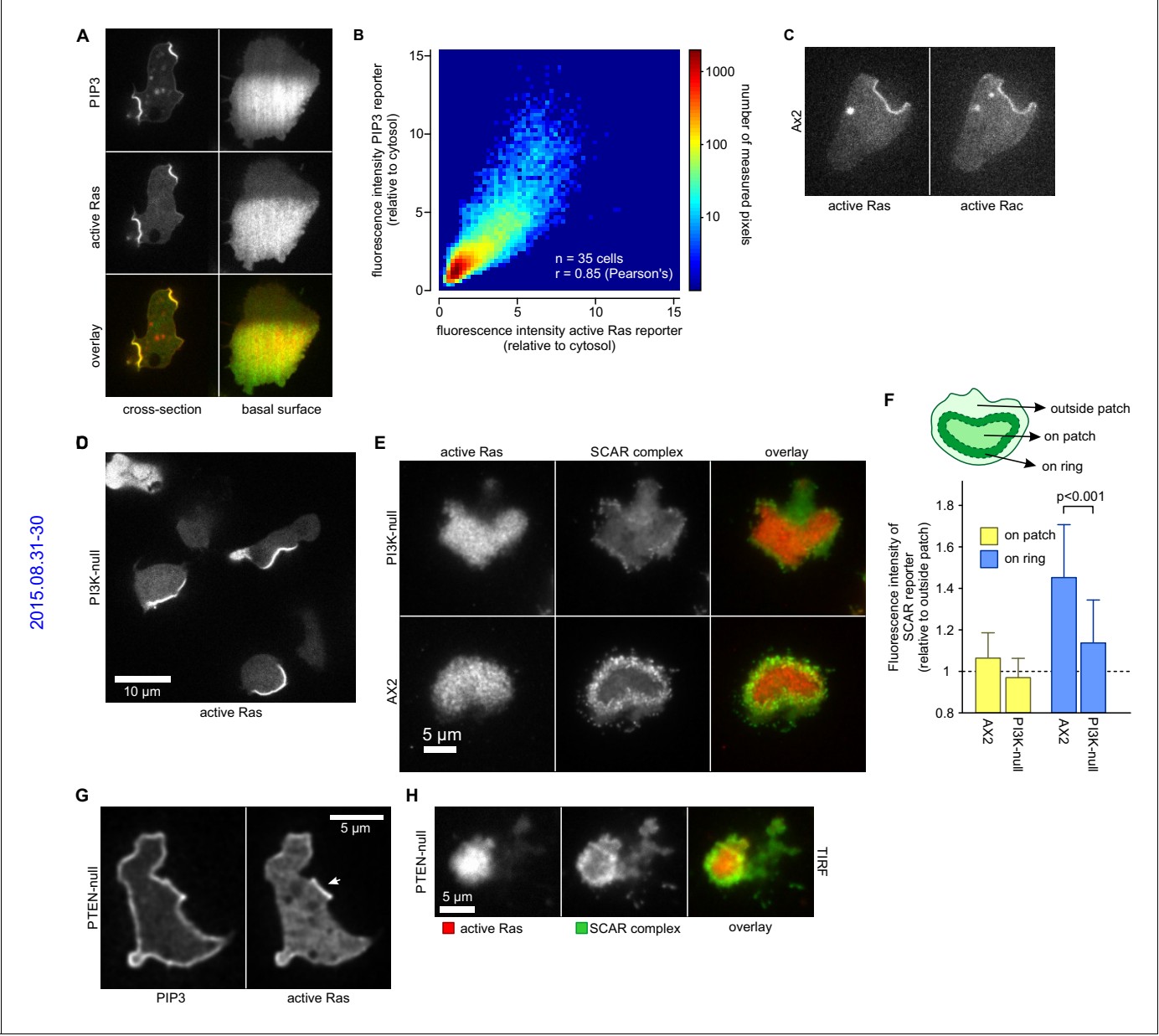

**Figure 6.** PIP$_3$ patches are supported by coincident patches of activated Ras, which can recruit weaker SCAR rings. (**A**, **B**) PIP$_3$ and activated Ras domains are essentially coincident in vegetative Ax2 cells: (**A**) dorsal patches (macropinocytic cups) and basal patches. Left and right panels show different cells; (**B**) 2-D histogram showing strong correlation between PIP$_3$ and activated Ras reporters. Fluorescence intensity values of the active Ras reporter and PIP$_3$ reporter along the perimeter of 35 cells are plotted against each other. (**C**) Macropinocytic signal patches additionally coincide with patches of active Rac. Shown is a representative image of a macropinocytic cup in a vegetative Ax2 cell co-expressing a marker for active Ras (Raf1-RBD) and active Rac (pakB-CRIB). (**D**, **E**) mutant lacking all Ras-activated PI3-kinases (strain HM1200) still forms Ras patches, both off the substratum (**D**) and basally- see (**E**). (**E**, **F**) Basal patches of the PI3-kinase mutant recruit SCAR to their periphery, though less strongly than the wild-type, Ax2 (TIRF images). Error bars indicate the standard deviation. Thus, loss of PI3K signalling does not allow SCAR to trespass on the Ras patch. (**G**, **H**) Ras patches (indicated by white arrow) remain discrete, despite globally high levels of PIP$_3$ in PTEN-null cells (strain HM1289) and these domains still invariably recruit a complete SCAR ring to their edges.

(*Postma et al., 2003, 2004*). Our results argue against an essential role for feedback from PIP$_3$ to Ras, because activated Ras patches can form independently of type-1 PI3-kinases and are still able to recruit SCAR to their edges, albeit less efficiently than when PI3-kinases are present. Thus it

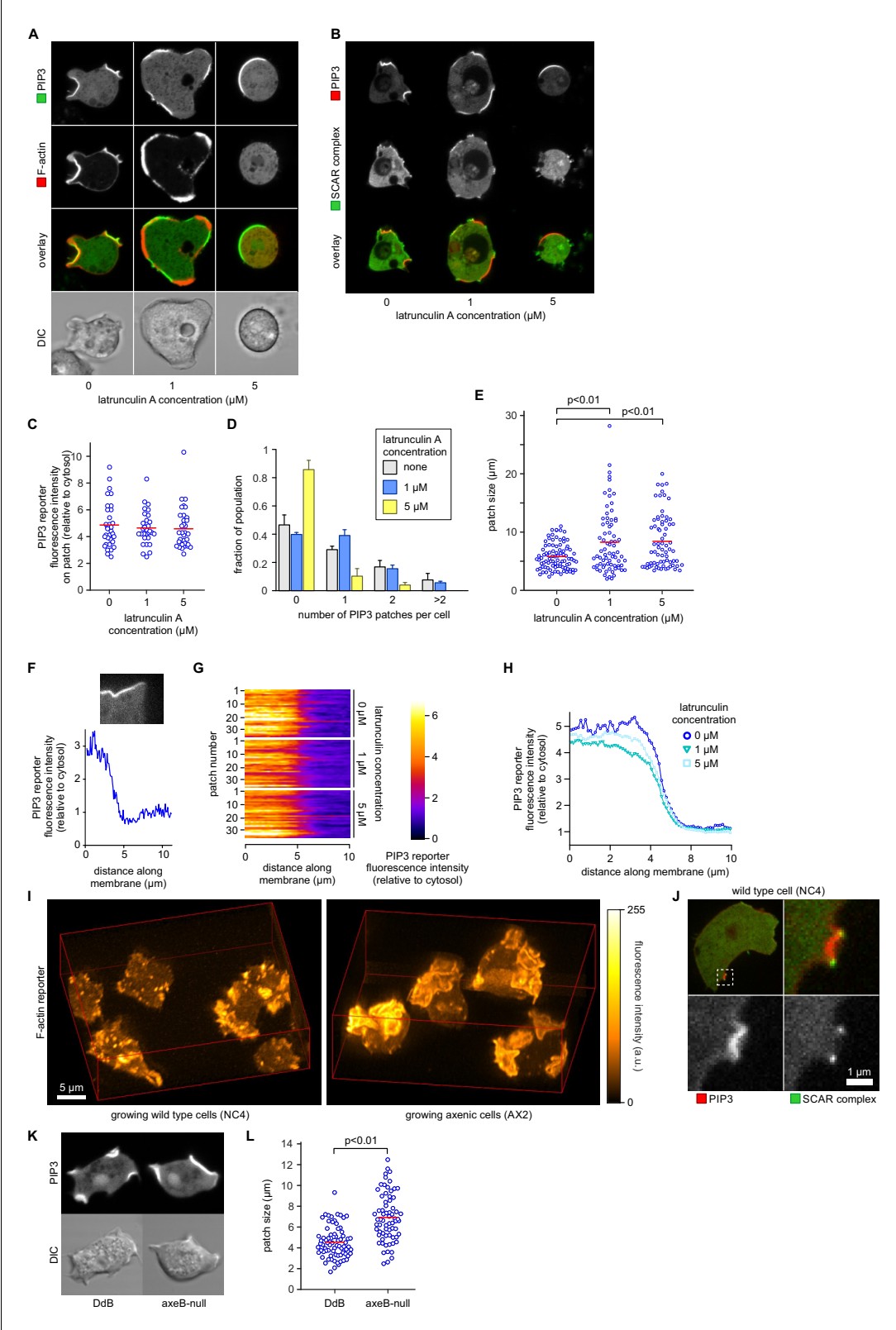

**Figure 7.** PIP$_3$ patch formation does not require an encircling actin ruffle, but patch size is regulated by Ras activity. (**A**) The actin cytoskeleton is depolymerized by treatment with latrunculin-A and ruffles suppressed. Vegetative cells expressing an F-actin marker (Lifeact) and a PIP$_3$ marker (PH-CRAC) were treated for 15 min with the indicated amounts of latrunculin-A. The F-actin marker is undetectable in the cortex when treated with 5 µM latrunculin-A. (**B**) PIP$_3$ patches remain after ruffles are suppressed by depolymerisation of the actin cytoskeleton. SCAR remains associated with the

*Figure 7 continued on next page*

*Figure 7 continued*

patch edge under 1 µM latrunculin-A, but is lost when treated with 5 µM latrunculin-A. (C–E) Ruffles are not essential for PIP$_3$ patch formation. Treatment with latrunculin-A does not significantly change PIP$_3$ patch intensity, but leads to an increase in size and a decrease in number. (F–H) Patch boundaries are sharply defined and this does not depend on an enclosing ruffle. (F) Membrane fluorescence intensity across the edge of PIP$_3$ patches was measured (G) Measured intensity profiles along the edge of 36 patches of both treated and untreated cells, with each line representing a single patch. (H) Mean fluorescence intensity of the PIP$_3$ reporter along the patch edges of treated and untreated cells, obtained by averaging the profiles in the previous panel. (I) Macropinocytic cups in vegetative cells of wild-type cells are smaller than those of axenic strains. Shown is a maximum intensity projection of the F-actin reporter LimEΔcoil of a field of vegetative cells, recorded using lattice light sheet microscopy. (J) SCAR is still recruited to the edges of the PIP$_3$ patch of the small macropinocytic cups of wild-type cells. (K–L) Increased Ras activity leads to larger PIP$_3$ patches and macropinocytic cups. Ras activity was increased by knock-out of the RasGAP, NF1 (*axeB* is the gene encoding NF1). Parental DdB and knock-out cells were cultivated for 48 hr in axenic medium to maximally up-regulate macropinocytosis. (K) Confocal image of macropinocytic patches in wild-type DdB and *axeB* null cells; (L) Quantification of the patch size in both cell types. Loss of NF1 leads to a significant increase in macropinocytic patch size.

The following figure supplement is available for figure 7:

**Figure supplement 1.** Basal PIP$_3$ patches (actin waves) are absent from wild-type cells.

appears that the kinetics that lead to patch formation must lie largely within the compass of the GEFs and GAPs activating and inactivating Ras.

We can only speculate on how SCAR is recruited to the periphery of Ras/PIP$_3$ patches. One possibility is that SCAR and Arp2/3 are preferentially recruited by newly synthesised F-actin (*Ichetovkin et al., 2002*) produced by formins (*Jasnin et al., 2016*), which might therefore be the initial actin nucleator to be recruited. However, this seems unlikely in the light of recent work showing that ForG contributes to the base of the macropinocytic cup, but seemingly not to the extending lip (*Junemann et al., 2016*). Alternatively, SCAR might be moved to the periphery of Ras/PIP$_3$ patches, perhaps by myosin-1 motors. Early work showed that myosin-1 is genetically important for macropinocytosis in *Dictyostelium* (*Novak et al., 1995*; *Titus, 2000*). In support of the genetic evidence, myosin-1 isoforms are recruited to macropinocytic cups in both *Dictyostelium* and *Acanthamoeba*

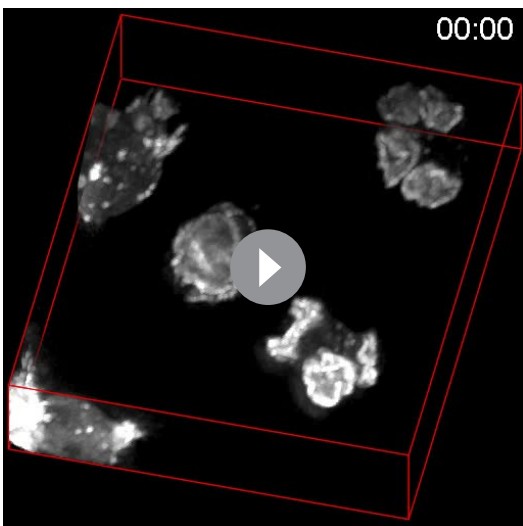

**Video 13.** The morphology of vegetative cells from axenic strains is dominated by large circular ruffles. Shown is a maximum intensity projection of the fluorescence intensity of the F-actin marker LimEΔcoil (Jitter in this movie was due to technical issues with the microscope's Z-drive). Images were taken on a lattice light sheet microscope.

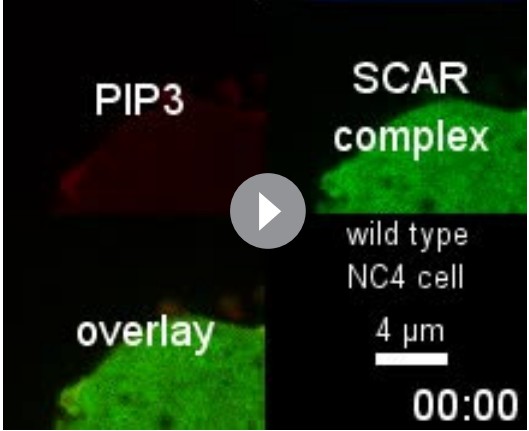

**Video 14.** SCAR follows PIP$_3$ patches in macropinocytic cups of wild-type NC4 cells. Detail of a vegetative cell from the wild strain NC4 expressing a marker for PIP$_3$ and a marker for the SCAR complex. Macropinocytosis is rapid and small but the cups still show SCAR puncta on their edges. Images were taken on a spinning disk microscope. It should be noted that due to the small size of the macropinocytic cups, the top and bottom of the cup are frequently in the focal plane, resulting in an overlap in both signals.

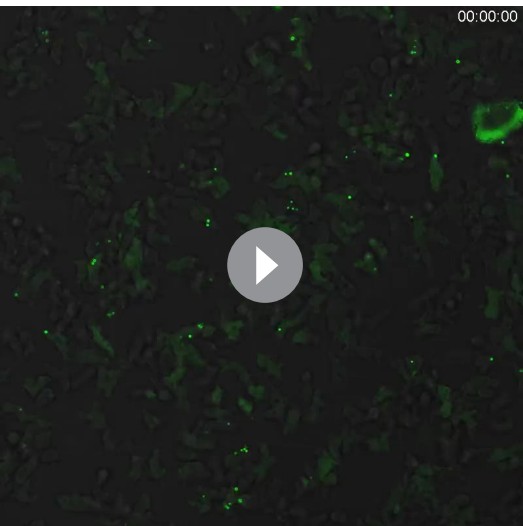
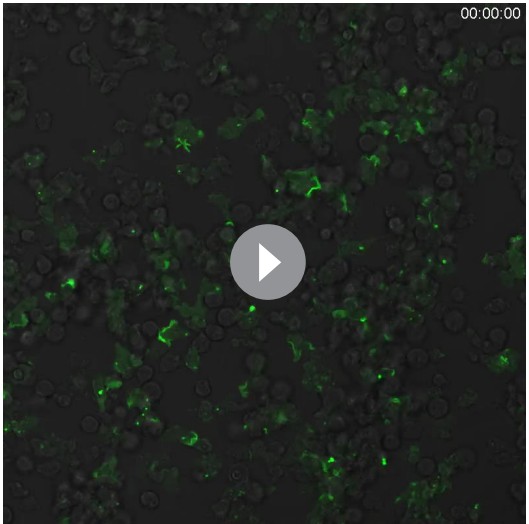

**Video 15.** Basal patches are dominant in axenic cells during early starvation. Cells from the axenic strain Ax2 were washed free of nutrients and left to develop autonomously under non-nutrient buffer. Time-lapse images were taken using a confocal microscope. Z-plane was set so that the basal membrane of the cell was in focus. Shown is an overlay of the fluorescence signal of a PIP$_3$ marker with the trans-illuminated image.

**Video 16.** Basal patches are absent from non-axenic wild type cells. Cells of the wild-type strain NC4 were washed free of nutrients and left to develop autonomously under non-nutrient buffer. Time-lapse images were taken using a confocal microscope. Z-plane was set so that the basal membrane of the cell was in focus. Shown is an overlay of the fluorescence signal of a PIP$_3$ marker with the trans-illuminated image.

(*Brzeska et al., 2012*; *Ostap et al., 2003*), most likely due to their affinity for PIP3 (*Chen et al., 2012*). The PIP3-binding MyoE and MyoF are recruited in the centre and MyoB at the periphery of macropinocytic cups, forming a striking 'bull's eye' pattern (*Brzeska et al., 2016*; *Dieckmann et al., 2010*). In such a scenario, CARMIL may provide the link between myosin-1 and SCAR (*Jung et al., 2001*).

Our work also has implications more specific to *Dictyostelium* biology. First, the basal actin waves, which give a valuable window into actin dynamics (*Bretschneider et al., 2004*, *2009*; *Gerisch, 2010*), appear to be formed as a consequence of the loss of NF1 in standard laboratory axenic strains. Knowing this should allow for better manipulation of these waves and for modelling to take account of their underlying need for activated Ras (*Arai et al., 2010*; *Khamviwath et al., 2013*; *Sasaki et al., 2007*; *Taniguchi et al., 2013*). Second, we consider that all patches of PIP$_3$ and activated Ras are related by their common recruitment of SCAR to their periphery and are therefore likely to organise circular rings of actin polymerization, rather than the solid blocks characteristic of pseudopods. Therefore, the proposed role of these patches in chemotaxis, where they have been mistaken for pseudopods, needs to be re-evaluated.

In summary, our work suggests a general hypothesis for the formation of cupped actin structures: that these structures arise from a ring of actin polymerization formed by recruiting actin nucleators to the periphery, but not the centre, of self-organizing patches of intense Ras and PIP$_3$ signalling. This hypothesis suggests many new lines of experimentation.

## Materials and methods

### Cell strains, cultivation and fluid uptake assay

The following *Dictyostelium discoideum* strains were used: Ax2 (R. Kay lab strain), NC4 (from K. Raper, obtained via P. Schaap), DdB (from M. Sussman, obtained via D. Welker), NC66.2 (from D. Francis), Ax3 (R. Chisholm laboratory strain, obtained via Stock Center) and Ax4 (W. Loomis

laboratory strain, obtained via Stock Center). Axenic strains were cultured in Petri dishes under HL5 medium (Formedium, Hunstanton, UK) using standard methods. Non-axenic strains were cultivated on SM agar plates with a lawn of live *Klebsiella pneumoniae* and where necessary washed free of bacteria by repeated low-speed centrifugation from KK2 (20 mM $KH_2PO_4/K_2HPO_4$, 2 mM $MgSO_4$, 0.1 mM $CaCl_2$, pH 6.2) (for detailed protocols of these standard techniques, see (*Kay, 1987*) and dictybase.org/techniques/). Mutant strains, all in the parental Ax2 (Kay) background, are listed in Supplementary Material, *Table 1*.

Fluid uptake was measured using TRITC-dextran and flow cytometry. Cells were grown on bacterial lawns, washed free of bacteria, resuspended in HL5 with antibiotics, 50 µl aliquots were distributed into 96 plates and allowed to adapt for about 18 hr, until macropinocytosis was maximally upregulated. TRITC-dextran was added to 0.5 mg/ml in HL5 to the wells and the cells incubated for various times, after which the TRITC dextran was removed, the cells washed once and uptake terminated with ice-cold, 5 mM $NaN_3$, which also detaches the cells. Fluorescence in individual cells was then measured by flow cytometry, and the rate determined while uptake was linear with time (first 45–60 min).

## DNA constructs and transfection

Single and dual expression vectors were used for all experiments (*Veltman et al., 2009*). Specifically, the following vectors were used: plasmid pDM1219 - expression of mCherry-LimE∆coil (residue 1–145 of Dd LimE), pDM767 - dual expression of HSPC300-GFP and PH-CRAC-mRFPmars (residue 1–126 of Dd DagA), pDM1492 - dual expression of mCherry-RBD-Raf1 (residue 1–134 of Hs Raf1) and PH-PkgE-mCherry (residue 1–100 of Dd PkgE), pDM1383 - dual expression of HSPC300-GFP and mCherry-RBD-Raf1 and pDM1424 - dual expression of HSPC300-GFP and PH-PkgE-mCherry.

The act6 promoter that drives the resistance marker on the expression vectors is not active when cells are cultivated using bacteria as a food source. Therefore, this promoter was replaced by the *coaA* promoter (bp −293 to bp −1 relative to the start codon of *coaA*) for those vectors that were used to transfect non-axenic, wild-type cells.

Transfection of non-axenic cells was performed as follows: $5 \times 10^6$ cells were harvested from the feeding front of an SM agar plate, washed once in H40 buffer (40 mM HEPES/KOH pH 7.0, 1 mM $MgCl_2$), and resuspended in 100 µl H40 buffer. Cells were mixed with 5 µl miniprep DNA (~0.5–1 µg total) and put on ice. Cells were then electroporated with two square waves of 350 V, 8 ms, 1 s apart using a Gene Pulser Xcell (Biorad) and immediately transferred to a Petri dish with SorMC buffer (15 mM $KH_2PO_4$, 2 mM $Na_2HPO_4$, 50 µM $MgCl_2$, 50 µM $CaCl_2$, pH 6.0) supplemented with live *Klebsiella pneumoniae* at an $OD_{600}$ of 2. Selection marker was added after 5 hr (10 µg/ml G418 or 100 µg/ml hygromycin).

## Image acquisition

Lattice light sheet microscopy 3D images were acquired as described (*Chen et al., 2014*), using a massively parallel array of coherently interfering beams comprising a non-diffracting 2D optical lattice as light sheet illumination focused by 0.65 NA objective for excitation (Special Optics). This creates a coherent structured light sheet that can be dithered to create uniform excitation in a 400 nm thick plane across the entire field of view determined by the length of the light sheet. In order to obtain the array of lattice light sheet, a binary spatial light modulator (SXGA-3DM, Forth Dimension Displays) is placed conjugate to the sample plane, and a binarized version of the desired structured pattern at the sample is projected on the display. In the time-lapse dithered mode, 3D stacks were acquired either by moving the detection objective (Nikon, CFI Apo LWD 25XW, 1.1 NA, 2 mm WD), which is synchronized with the scanning galvo mirror, or moving the sample by fast piezoelectric flexure stage (Physik Instrumente, P-621.1CD) with 100 ~150 z planes, to have about 20 µm in z axis with respect to the detection objective. Exposure time was 5 or 10 msec per plane, for a total exposure time of ~1 s for one 3D stack and a 1 s pause was added between each time point to have time series data. Raw data was deconvolved via a 3D iterative Lucy-Richardson algorithm in Matlab (The Mathworks, Natick, MA) utilizing an experimentally measured point spread function.

Spinning disk microscopy was performed on an Andor Revolution system with a Yokogawa CSU spinning disk confocal unit. The microscope was fitted with a 1.49 Plan Apo 100x oil immersion objective and an additional 1.2x magnification lens. GFP and mCherry signals were separated by a

Tucam beam splitter and detected using two Andor iXon Ultra backlit EMCCD cameras with 16 μm pixel size. Z-scans were performed with the 1.5x optovar in place using 70 ms exposure per frame and a Z-spacing of 0.19 μm. Typically, 80 frames were collected from each camera in a total of 8 s.

TIRF microscopy was performed using a Nikon N-STORM system fitted with a 1.49 Plan Apo 100x oil immersion objective and the 1.5x optovar in place. GFP and mCherry fluorescence signals were recorded sequentially on an Andor iXon Ultra backlit EMCCD camera.

Confocal microscopy was performed on a Leica SP8 system using a 1.4 NA plan apo oil immersion objective and GFP/mCherry fluorescence was detected using two HyD detectors. All microscopy was performed at room temperature.

## Image analysis

General image handling, such as brightness/contrast adjustments and generation of kymographs was done using ImageJ (NIH). 3D cellular fluorescence images were generated as follows. A Z-stack was recorded on a spinning disk microscope using previously indicated settings. The dataset was deconvolved with Huygens Professional software (Scientific Volume Imaging) using a calculated point spread function. The images presented are maximum intensity projections of the deconvolved dataset.

Correlation between speed and membrane fluorescence intensity was analysed using Quimp11 (www.warwick.ac.uk/quimp). Identification of membrane pixels and measuring their fluorescence intensity was done using a custom-written MATLAB (The MathWorks) script (*Supplementary file 1* and *Source code 1*).

Image sets that were used for quantification were taken from at least two independent transfections. Only those cells with very low HSPC300-GFP expression were included for analysis, as overexpression dramatically reduces image contrast. For the quantification of SCAR fluorescence on the edge and centre of macropinocytic cups a paired 2-tailed T-test was used in *Figure 3B and D* and a 2-tailed T-test was used in *Figure 6F*.

All lattice light sheet microscopy movies (1–7 and 12–13) show a maximum intensity projection of the fluorescence intensity. The F-actin marker LimEΔcoil is used in all images unless otherwise specified. Images were deconvolved using a custom-written Richardson-Lucy algorithm. The maximum intensity projection was generated using Huygens software. Indicated time is in the min:sec format.

## Acknowledgements

We wish to thank Sean Munro for comments on the manuscript and the Biotechnology and Biological Sciences Research Council (Grant number BB/K009699/1 to RRK and DV), Medical Research Council (reference number U105115237 to RRK) for support.

## Additional information

### Funding

| Funder | Grant reference number | Author |
|---|---|---|
| Biotechnology and Biological Sciences Research Council | BB/K009699/1 | Douwe M Veltman Robert R Kay |
| Medical Research Council | U105115237 | Robert R Kay |

The funders had no role in study design, data collection and interpretation, or the decision to submit the work for publication.

### Author contributions

DMV, Conception and design, Acquisition of data, Analysis and interpretation of data, Drafting or revising the article; TDW, Acquisition of data, Analysis and interpretation of data; GB, Analysis and interpretation of data, Contributed unpublished essential data or reagents; B-CC, Acquisition of data, Drafting or revising the article; EB, Acquisition of data, Contributed unpublished essential data or reagents; RHI, Conception and design, Drafting or revising the article; RRK, Conception and design, Analysis and interpretation of data, Drafting or revising the article

Author ORCIDs

Douwe M Veltman, http://orcid.org/0000-0002-9512-3235

## Additional files

**Supplementary files**

• Supplementary file 1. Manual for the track_membrane MATLAB script.

• Source code 1. Track membrane MATLAB script.

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
