## [Decision Letter]

[Editors’ note: a previous version of this study was rejected after peer review, but the authors submitted for reconsideration. The first decision letter after peer review is shown below.]

Thank you for submitting your work entitled "A plasma membrane template for macropinocytic cups" for consideration by *eLife*. Your article has been evaluated by Vivek Malhotra (Senior Editor) and three reviewers, one of whom is a member of our Board of Reviewing Editors. Our decision has been reached after consultation amongst the reviewers. Based on these discussions and the individual reviews pasted below, we regret to inform you that your work will not be considered further for publication in *eLife*.

Dr. Joel Swanson has agreed to reveal his identity as one of the peer reviewers of your paper.

All in all, the reviewers were satisfied with the images, the high quality of data and your interesting hypothesis. However, they were concerned that your conclusions were not sufficiently justified by the data especially with respect to the mechanism of ring formation and the actual function of the ring in cup formation. We feel their major concerns cannot be addressed within the usually permitted period of 2-3 months. But, given the potential importance of the findings, we would be willing to consider a new manuscript for review by the same reviewers, provided you can address the reviewers’ concerns. Otherwise, we strongly urge you to submit this work elsewhere.

*Reviewer #1:*

This paper investigates the mechanisms driving the formation of macropinocytic cups in *Dictyostelium*. Using high end imaging (lattice sheet microscopy), the authors uncover that the actin nucleation promoting factor SCAR accumulates in a thin ring at the edge of PIP_3_ patches. They then explore PIP_3_ patches in a variety of other cellular processes, such as phagocytosis and basal actin waves, and demonstrate that similar SCAR rings are present. Finally, the authors investigate the role of Ras in the formation of PIP_3_ patches.

The imaging is beautiful and the results do suggest a morphogenetic mechanism. However, the work remains at a rather preliminary level and does not go much beyond the observation of the SCAR rings. Many open questions remain: What is the mechanism of SCAR ring formation? The authors discuss the possible involvement of myosin 1, could they explore this experimentally?

If the SCAR ring is indeed responsible for the curving of the membrane and the formation of a circular actin ruffle, why do all the structures with such a ring not adopt a similar geometry? How does the morphogenetic mechanism suggested by the authors actually work? It would have been useful to follow the dynamics of SCAR during macropinosome formation. Is the PIP_3_ patch initially flat and then the membrane curves out? And if so, does this event correspond with SCAR ring formation? Or does the ring initially promote patch expansion, like in an actin wave, and if so, how does the transition to outwards polymerisation occur? Addressing any aspect of the mechanism of ring formation or of its morphogenetic effect would bring the paper beyond the level of an interesting observation.

1) The authors use the correlation between HSPC300-GFP recruitment and forward extension of pseudopods as a proof that this construct localizes active SCAR. This is not entirely clear. Is SCAR itself (in an inactive form) still present at the edge of stalled pseudopods? And if not, could HSPC300 localize together with SCAR independently of its activity?

2) If CRAC PH-domain binds both PIP_3_ and PIP_2_ as stated by the authors (subsection “SCAR is recruited to the periphery of PIP_3_ domains in macropinocytic cups”, third paragraph), why do they consider it a readout of PIP_3_ localization?

3) Some of the figures (e.g. most of the data displayed on Figure 4; Figure 5) lack quantification, and while they display beautiful images, it would be helpful to quantify the readouts, or at least mention in the figure legend of how many cells and experiments the presented image is representative.

4) Figure 4 is interesting but raises several questions: why has WASP been localized only for basal actin waves? Does WASP also follow the same trend as SCAR in the other cases of PIP_3_ patches? In panel H, SCAR appears to form larger domains that overlap with PIP_3_ to some extent. How does this fit with the other observations?

5) It is not clear how the measurements displayed in panel 5E were actually done: is the background measured outside the cell? And if so, is this the right normalization? And if not, where would the background region be in pictures like those displayed in panel D? Also, the authors claim that SCAR recruitment is less strong in PI3K-null cells than in wt, however, the fluorescence at center also seems lower in this case. It would be better to plot the ratio of center to edges (both measured in the same patch), which would also avoid the use of the background signal.

6) The authors discuss the possible presence of a diffusion barrier at the edge of Ras/PIP_3_ domains, could they discuss what would and actin based barrier (as suggested) prevent from diffusing? How would such a barrier work?

*Reviewer #2:*

The manuscript by Veltman, et al., examines the dynamics of F-actin, PIP_3_ Ras-binding domains (RBD) and SCAR/WAVE during macropinosome formation and related movements in *Dictyostelium*. Using state-of-the-art fluorescence microscopy, the authors demonstrate a distinct spatial arrangement of PIP_3_ RBD and SCAR in forming macropinocytic cups, with PIP_3_ and RBD localizing to patches of plasma membrane within cups and SCAR (inferred by imaging the SCAR component HSPC300-GFP) localizing to cup rims, at the edges of ruffles that define PIP_3_ patches. The images provide a striking correlation between the localization of PIP_3_ and RBD patches and of SCAR at the edges of ruffles that delineate those patches. They show further that similar patterns can be detected in wild-type amoebas which make macropinosomes more infrequently than those which are adapted to broth culture.

My major concern is with the statements, in the Abstract and elsewhere, that "patches of intense PIP_3_ signalling […] recruit SCAR/WAVE complex to their edges, […] driving a ring of actin polymerization that forms the walls of the cups" (Abstract). This implies that PIP_3_ patches precede the SCAR localization to patch edges, which is not demonstrated by the experiments. The figures provided, especially Figure 4, indicate that PIP_3_ patches form coincident with SCAR concentration at leading edges. Instead, I would interpret the images as supporting a model in which SCAR defines a boundary, perhaps a diffusion barrier, that permits amplification of PI 3-kinase activity within a circular domain in the inner leaflet of plasma membrane; that is, SCAR either prescribes the PIP_3_ patch or assembles coincidently into the concentric arrangement. This essential claim of the manuscript about which comes first could be addressed by experiments which measure the timing of PIP_3_ patch formation relative to that of SCAR localization and ring formation.

Secondly, mechanistic interpretations (i.e., PIP_3_ patches recruit SCAR) would require more than the correlative evidence provided here, such as localization after experimental manipulations of the system components. Some mechanism is suggested by the experiments of Figure 5, which show RBD patches forming without corresponding patches of PIP_3_, and which suggest that PIP_3_ patches do not prescribe SCAR rings. However, the mechanism implied by this result remains speculative.

*Reviewer #3:*

In this manuscript the authors study the organization of macropinocytic cups in *Dictyostelium* cells. They first characterize the dynamic structure of the macropinocytic cups in 3D using the new lattice light sheet microscopy technology. They also compare and contrast the macropinocytic structures to basal actin waves and to pseudopodia. The authors show that active SCAR protein localizes to the rim of the macropinocytic cups. The SCAR protein, and to a lesser extent WASP, follows the edge of a membrane patch enriched in PIP_3_ and Ras activity. The localization of SCAR to the edge of PIP_3_ patches is shown to be very robust as it is seen under various growth conditions and genetic backgrounds. The authors suggest that the SCAR pattern explains the formation of the cup shaped actin structures driving the formation of the macropinosomes. Finally, the authors show using mutant cells that PIP_3_, may contribute, but is not required for the formation of the SCAR pattern.

The finding that SCAR localizes to the rim of PIP_3_ patches is interesting and likely to be important for mechanistic understanding of the actin driven formation of the macropinosomes. However, the molecular mechanism that forms the SCAR pattern around the PIP_3_ patches remains unknown. Several hypotheses are discussed, but not tested in the manuscript. Testing these models may be beyond the scope of this manuscript.

The manuscript is clearly written, and the presented data is very beautiful, quantitative and of high technical quality. The key conclusions of the manuscript are well supported by the data.

I would, however, suggest toning down some overstated conclusions that might be misleading for some readers.

1) At the end of the subsection “All PIP_3_ patches, whatever their origin, recruit SCAR to their periphery” the authors state that: "[…] any process that causes patches of PIP_3_ will cause recruitment of SCAR at the edges […]". This is a rather exaggerated conclusion based on colocalization data, especially when the authors later show that PIP_3_ is not needed for SCAR localization.

2) At the end of the subsection “Macropinosomes in wild-type cells show the same as organisation as in axenic cells, but basal PIP_3_ waves do not form” they also conclude "[…] that PIP_3_ patches cause peripheral SCAR and actin rings to form […]". Again this is too strong a conclusion. No evidence is presented that PIP_3_ causes the SCAR localization.

3) In the first paragraph of the Discussion it says: "This work offers an explanation how the architecture of these rings is controlled." The manuscript describes the SCAR ring architecture, but doesn't really explain the mechanisms that control it.

[Editors’ note: what now follows is the decision letter after the authors submitted for further consideration.]

Thank you for resubmitting your work entitled "A plasma membrane template for macropinocytic cups" for further consideration at *eLife*. Your revised article has been favorably evaluated by Randy Schekman (Senior editor) and three reviewers, one of whom is a member of our Board of Reviewing Editors.

The manuscript has been improved but there are some remaining issues that need to be addressed before acceptance. No additional experimental work is needed, but the Discussion section should be revised to clarify points identified by reviewers 2 and 3.

1) "The authors should address what is known about the role of specific Racs that have been described as having a role in macropinocytosis, such as Rac1A and Rac1C and perhaps RacB, so as to remove some of the ambiguity from the presentation of the data and synthesis of the results. "

2) Known roles for RasC or RasG in macropinocytosis, if any, should be mentioned.

3) Known roles for myosin 1 isoforms in macropinocytosis should be mentioned.

4) Citations of Hoeller et al. and Seastone et al. should be included with the reference to Clark et al. 2014.

5) Additional thoughts about the nature of the diffusion barrier that remains after latrunculin treatment are optional but encouraged.

*Reviewer #1:*

This manuscript analyzes the spatial and temporal organization of actin PIP_3_, Ras, Rac and SCAR during the formation of macropinosomes in *Dictyostelium*. It demonstrates that SCAR localizes to the edges of plasma membrane patches of PIP_3_ coincident with patch formation. The PIP_3_ patches, which also contain active Ras and Rac, prescribe the localization of SCAR and the accompanying actin polymerization that forms the cup. Additional experiments support a model in which the size of the PIP_3_ patch is prescribed by active Ras, and not vice versa; this is an important mechanistic insight. An alternative hypothesis, that the PIP_3_ patch is prescribed by the forming actin-rich cup, is excluded by experiments showing that PIP_3_ patches can form in cells whose actin has been depolymerized by latrunculin A.

The revised manuscript adequately addresses the critiques from the first reviews. The additional experiments provide mechanistic analyses that were missing from the first manuscript. Revised text addresses earlier concerns about interpretation of the results.

The title of the manuscript is acceptable. Supplementary videos provide important supporting data for the claims of the paper, especially regarding the relative distributions of PIP_3_ and SCAR.

*Reviewer #2:*

Cells can internalize large volumes of liquid via macropinocytosis, a dynamic process driven by actin polymerization. Veltman et al. beautifully describe macropinocytosis in the *Dictyostelium* amoebae. They show that macropinocytic cups do not arise from flat membrane protrusions or pseudopods as had been proposed but rather arise de novo from the membrane of a *Dictyostelium* cell. The Arp2/3 regulator SCAR is tightly localized to the rim of the extending cup, surrounding a membrane patch enriched for both active Ras and PIP_3_, where it is maintained all throughout its formation, expansion and closure. Cells with significantly reduced levels of PIP_3_ continue to make macropinocytic cups, with a core patch of activated Ras bordered by SCAR, establishing that activation of small G proteins is the initial driver of cup formation. Support for this comes from the finding that true wild type strains extend significantly fewer and smaller macropinocytic cups, however upon disruption of the NF1 RasGAP (which is mutated in axenic laboratory strains) macropinocytosis is dramatically upregulated. Together with previous work in the field, the results establish that Ras/Rac activation is critical for macropinosome formation that promotes the localized production of PIP_3_ on the membrane and selectively recruits and/or activates SCAR at the border that, in turn activates Arp2/3 mediated actin polymerization to drive cup extension.

Activated Ras, PIP_3_, and SCAR are known to have significant roles in macropinocytosis, but it has not been clear which is the primarily responsible for initiating cup formation. Here the authors provide new insights into this problem by showing that Ras/Rac activation at the membrane corresponds with the recruitment of SCAR to the periphery, likely playing critical role in initiating a ring of actin polymerization surrounding the patch at the membrane. PIP_3_ appears to be dispensable for this initial step but must play a role in subsequent steps. The results also demonstrate that membrane patches of PIP_3_ are not associated with pseudopods but are rather mainly involved in the formation of macropinocytic structures, an important clarification for the field that has focused a good deal of attention to the role of PIP_3_ in directed migration. Naturally, these findings raise the critical question of how specific small GTPases with roles in macropinocytosis act together with PI3K to orchestrate subsequent events. Activated Ras/Rac is permissive for SCAR activation at the edges of the forming and mature cup but how the relatively large region of active Ras on the membrane causes such restrictive localization of SCAR remains an intriguing mystery.

Specific comment:

Reporters for activated Ras and Rac are localized to the macropinocytic cup and a number of small GTPases, such as RasC and RasG, are implicated in macropinocytosis, as previously shown by others and also described here. The authors should address what is known about the role of specific Racs that have been described as having a role in macropinocytosis, such as Rac1A and Rac1C and perhaps RacB, so as to remove some of the ambiguity from the presentation of the data and synthesis of the results.

Have the authors examined either the RasC or RasG null mutant to see what step in macropinocytosis either of these small GTPases affects – macropinosome formation or cup size/extension? The addition of such data could be an important first step towards clarifying the role of specific small GTPases in initiating macropinocytosis.

*Reviewer #3:*

This study dissects the spatial and temporal patterns of PIP_3_, SCAR and actin in the genesis of macropinocytic lamellae and cups with unprecedented precision. The study integrates, extends and explains previous studies on the generation, propagation and function of actin waves at the ventral surface of the cell, by showing their relationship with ruffles and cups that give rise to macropinosomes on the dorsal surface.

One of the major claims is that PI3K does not play an instructive role in the generation of cups, but instead demonstrates such a role for Ras and not for Rac.

Another extremely important discovery is that the RasGAP ortholog of NF1, which absence or lack of functionality is determinant for the capacity of *Dictyostelium* cells to live from fluid phase uptake, dictates the size of the cups.

Overall, this revised version of the manuscript presents very strong experimental evidence for the claims, is richly documented both by time-lapse microscopy with unprecedented 3D and temporal resolution, and exquisite reconstructions and analyses. All important results are quantified and analysed with appropriate statistical tools.

It is also worth noting that I was not among the reviewers of the original manuscript, but am fully satisfied by the answers given to all criticisms, mainly in the rich and complete additional experimental evidence as well as in the reformulation of the claims and the overall more objective tone of the text.

Therefore, I do not have strong criticisms, but only a few points for which I would encourage the authors to bring some additional clarification and discussion.

1) Every PIP_3_ patch has a rim of SCAR, but it is not really clear to me whether every SCAR-positive structure (dot/patch/ring) has a core of underlying PIP_3_. The same can potentially be asked for combinations of activated Ras or Rac.

2) The claim that the LatA experiments rule out that a diffusion barrier is required to restrict the PIP_3_ patch is too general. It shows that an actin-dependent diffusion barrier is not required. But one might envisage other membrane heterogeneities that could serve as diffusion barrier (Lo versus Ld phases, rafts etc.).

3) Grinstein and colleagues had shown that the phagocytic cup lip serves as diffusion barrier, possibly implying that the geometry - the curvature - was the mechanism. The findings here with LatA-treated spherical cells refute this hypothesis. This should be mentioned.

4) Maybe a slightly expanded discussion on the possible roles of class I myosins would be cool. They were implicated in the ventral actin waves; they are known to participate in the recruitment and activation of the Arp2/3 complex; the absence of MyoK induces morphological similarities with the "smooth bell-shaped cell" morphology induced by a wave spreading through the cell footprint; absence of many class I myosins result in macropinocytic but also phagocytic and trafficking phenotypes.

---

## [Author Response]

[Editors’ note: the author responses to the first round of peer review follow.]

[…] *Reviewer #1:*

*This paper investigates the mechanisms driving the formation of macropinocytic cups in Dictyostelium. Using high end imaging (lattice sheet microscopy), the authors uncover that the actin nucleation promoting factor SCAR accumulates in a thin ring at the edge of PIP_3_ patches. They then explore PIP_3_ patches in a variety of other cellular processes, such as phagocytosis and basal actin waves, and demonstrate that similar SCAR rings are present. Finally, the authors investigate the role of Ras in the formation of PIP_3_ patches.*

*The imaging is beautiful and the results do suggest a morphogenetic mechanism. However, the work remains at a rather preliminary level and does not go much beyond the observation of the SCAR rings. Many open questions remain: What is the mechanism of SCAR ring formation? The authors discuss the possible involvement of myosin 1, could they explore this experimentally?*

We agree that myosin 1 is likely to be important in the formation of macropinocytic cups and are actively working in this area. We have made reporters to six different myo‐1 proteins and confirmed the bull’s eye pattern of recruitment to macropinocytic cups recently reported by the Korn lab (Cytoskeleton, 2016); we have also made single mutants in all these genes and some doubles and a triple, again confirming that they are important for macropinocytosis. However, this is a long‐term project, and unfortunately we are not in a position to add anything to the current paper from this work.

*If the SCAR ring is indeed responsible for the curving of the membrane and the formation of a circular actin ruffle, why do all the structures with such a ring not adopt a similar geometry?*

The circular actin ruffle contains Arp2/3 cross linked F‐actin, which in *Dictyostelium* is catalysed exclusively by members of the WASP family (SCAR, WASP and WASH). Since SCAR is recruited to the circular actin ruffle and there are no other candidates, we conclude that SCAR must be responsible.

All structures with a SCAR ring adopt the same geometry, but negative curvature is not always clear in each individual images. (1) A 2D confocal section through the center of a macropinocytic cup typically shows negative curvature, but this is less pronounced in different confocal slices. (2) The negative curvature induced by the actin ring cannot always be maintained in the center of the cup. Macropinocytic cups in axenic cells are unusually large and are presumably physically challenging. The center of such large macropinocytic cups can be flat or can even have positive (outward) curvature with negative curvature limited to the edge, where the actin ring is.

Other structures with a SCAR ring are also always associated with negative curvature when this is physically permitted. This is true in phagosomes and cell‐cell contacts, but not in basal waves, which are constrained to be flat. In this case the F‐actin protrudes into the cell interior for several microns, rather than pushing the membrane out.

*How does the morphogenetic mechanism suggested by the authors actually work? It would have been useful to follow the dynamics of SCAR during macropinosome formation. Is the PIP3 patch initially flat and then the membrane curves out? And if so, does this event correspond with SCAR ring formation? Or does the ring initially promote patch expansion, like in an actin wave, and if so, how does the transition to outwards polymerisation occur? Addressing any aspect of the mechanism of ring formation or of its morphogenetic effect would bring the paper beyond the level of an interesting observation.*

These are all very important questions and we have explored them in depth in the revised manuscript (new Figure 4).

We show that PIP_3_ patches in all cases have SCAR along their periphery. We did not identify any patches, even those in initiating macropinocytic cups, where a ring of SCAR/F‐actin was absent. Macropinocytic cups therefore do not initiate by a belated recruitment of SCAR along initially flat PIP_3_ patches.

We have also recorded time lapse images of PIP_3_ and SCAR during developing macropinocytic cups. During all stages of the macropinocytic cup lifetime, SCAR remains tightly associated to the edges of the PIP_3_ patch. We thus find no evidence for a transition between macropinocytic cup expansion and macropinocytic cup closure. SCAR drives cup progression at all stages.

*1) The authors use the correlation between HSPC300-GFP recruitment and forward extension of pseudopods as a proof that this construct localizes active SCAR. This is not entirely clear. Is SCAR itself (in an inactive form) still present at the edge of stalled pseudopods? And if not, could HSPC300 localize together with SCAR independently of its activity?*

We may not have been sufficiently clear in the text. We do not assert that HSPC300 specifically marks active SCAR complex. HSPC300 and SCAR form a stable complex (together with PIR121, NAP and Abi) and HSPC300 thus marks all SCAR complex. We assert that the SCAR complex is active at places where it strongly accumulates. All membrane areas with high concentrations of fluorescent SCAR complex show outward membrane movement, thus we assume that it is active at these positions.

In principle any of the subunits of the SCAR complex can be tagged with GFP and we have done so (Veltman et al. J Cell Biol. 2012 Aug 20; 198(4): 501–508.), but we prefer to tag HSPC300 due to its small size, facilitating experimental procedures.

We have revised the text to more concisely make the point regarding the SCAR reporter.

*2) If CRAC PH-domain binds both PIP_3_ and PIP_2_, as stated by the authors (subsection “SCAR is recruited to the periphery of PIP_3_ domains in macropinocytic cups”, third paragraph), why do they consider it a readout of PIP_3_ localization?*

PH‐CRAC is the conventional sensor for PIP_3_ in *Dictyostelium* and has been used for almost 20 years. We have used this marker so that our data can be more easily compared with previous work. Some *Dictyostelium* workers hypothesize that the PIP_3_ patches (marked by PH‐CRAC) recruit SCAR and result in pseudopod formation. Our work shows that this is not the case.

There are no truly PIP_3_‐specific probes available. All tested PIP_3_ probes, such as PH‐ Akt, PH‐Btk, PH‐Gab2 and PH‐CRAC strongly bind to PIP_3_, but also bind to 3,4‐PIP_2_ (though not 4,5‐PIP_2_) to a lesser degree (See for example Park et al. Mol Cell. 2008 May 9; 30(3): 381–392. or Manna et al. J. Biol. Chem. 2007, 282:32093‐32105). In all cases, the K_d_ for PIP_3_ is much lower than that for 3,4‐PIP_2_ and thus the probes are considered PIP_3_ probes. It should be noted that we tested various PIP_3_ probes and found no differences with respect to macropinocytic cups:

The probes do behave differently with regards to pseudopod formation and chemotaxis and we are currently preparing a manuscript to describe these differences.

*3) Some of the figures (e.g. most of the data displayed on Figure 4; Figure 5) lack quantification, and while they display beautiful images, it would be helpful to quantify the readouts, or at least mention in the figure legend of how many cells and experiments the presented image is representative.*

All images shown are representative. We should note here that a ring of SCAR around patches of Ras/PIP_3_ is detected in all cases, provided a microscope with a high NA objective is used (the SCAR puncta flanking the PIP_3_ patches are absolutely tiny) and provided that the SCAR reporter is not too highly overexpressed, in which case the high cytosolic background obscures the relatively weak fluorescence of the ring.

We have included in the text throughout the manuscript whether shown images are representative and added the number of analysed cells where appropriate.

*4) Figure 4 is interesting but raises several questions: why has WASP been localized only for basal actin waves? Does WASP also follow the same trend as SCAR in the other cases of PIP_3_ patches? In panel H, SCAR appears to form larger domains that overlap with PIP_3_ to some extent. How does this fit with the other observations?*

We included these images because we think that it is important to show that PIP_3_ patches do not recruit any of the Arp2/3 activating proteins to their centers. We show that WASP follows the same trend as SCAR in macropinocytic cups, but only very weakly (Figure 4, revised manuscript). Since the recruitment of WASP is not as strong as that of SCAR, the WASP signal was generally not sufficient to reliably detect it at the rim of PIP_3_ patches in 3D recordings (which are technically demanding) in all cases.

The referee is right to remark that in the uniform stimulation experiment (Figure 4) there are membrane areas where the PIP_3_ and SCAR signals partially overlap. In principle SCAR can thus be recruited transiently to areas with high levels of PIP_3_. However, this is specific to the (non‐physiological) uniform stimulation experiments and we assume that the duration of the SCAR recruitment to the membrane in these experiments is too short (only 2 seconds) for the PIP_3_ and SCAR to self‐organise into patches/rings.

*5) It is not clear how the measurements displayed in panel 5E were actually done: is the background measured outside the cell? And if so, is this the right normalization? And if not, where would the background region be in pictures like those displayed in panel D? Also, the authors claim that SCAR recruitment is less strong in PI3K-null cells than in wt, however, the fluorescence at center also seems lower in this case. It would be better to plot the ratio of center to edges (both measured in the same patch), which would also avoid the use of the background signal.*

The background signal is measured in a basal membrane region (revised Figure 6) where there is where there is no patch. In some cases the patch is large and there is not much area outside the patch. We disregarded images where the patch occupied the entire footprint of the cell and the background could not reliably be measured. In the shown images (revised Figure 6) the background was measured at the indicated areas.

The referee suggests measuring the ratio center:edges. We have not opted for this strategy as we argue that the signal at a location outside the patch is the best value to normalise by. The referee is correct to remark that the loss in SCAR recruitment will be smaller when normalising to the center of the patch. However, this does not affect our conclusion, which is that PIP_3_ is not required for SCAR ring formation.

We have replaced the word "background" by "outside patch" in the revised manuscript to more clearly indicate how the data is normalised.

*6) The authors discuss the possible presence of a diffusion barrier at the edge of Ras/PIP3 domains, could they discuss what would and actin based barrier (as suggested) prevent from diffusing? How would such a barrier work?*

The presence of a diffusion barrier was also suggested by reviewer 2 and we have addressed these questions in depth in the revised manuscript. See also the response to reviewer 2.

*Reviewer #2:*

*[…] My major concern is with the statements, in the Abstract and elsewhere, that "patches of intense PIP_3_ signalling [...] recruit SCAR/WAVE complex to their edges, […], driving a ring of actin polymerization that forms the walls of the cups" (Abstract). This implies that PIP_3_ patches precede the SCAR localization to patch edges, which is not demonstrated by the experiments. The figures provided, especially Figure 4, indicate that PIP_3_ patches form coincident with SCAR concentration at leading edges. Instead, I would interpret the images as supporting a model in which SCAR defines a boundary, perhaps a diffusion barrier, that permits amplification of PI 3-kinase activity within a circular domain in the inner leaflet of plasma membrane; that is, SCAR either prescribes the PIP_3_ patch or assembles coincidently into the concentric arrangement. This essential claim of the manuscript about which comes first could be addressed by experiments which measure the timing of PIP_3_ patch formation relative to that of SCAR localization and ring formation.*

We thank the reviewer for this clear exposition of an alternative model, which we have addressed with several new experiments, as described above.

We did not intend to imply any temporal order by using the word "template", i.e. that PIP_3_ patches precede SCAR rings. We intended to imply that the spatial arrangement of the ring is specified by that of the patch.

Our new data indeed indicates that SCAR appears simultaneously with Ras and PIP_3_ in macropinocytic cups (added Figure 1,Figure 4 and Video 4), agreeing with a spatial templating hypothesis, but not with temporal templating.

We have included new data on the temporal evolution to show that signals and F‐actin co‐evolve during macropinosome initiation. We no longer refer to the word "templating" throughout the text, to avoid possible confusion over who comes first and instead use the term "self organise".

*Secondly, mechanistic interpretations (i.e., PIP_3_ patches recruit SCAR) would require more than the correlative evidence provided here, such as localization after experimental manipulations of the system components. Some mechanism is suggested by the experiments of Figure 5, which show RBD patches forming without corresponding patches of PIP_3_, and which suggest that PIP_3_ patches do not prescribe SCAR rings. However, the mechanism implied by this result remains speculative.*

In the revised manuscript we have added proof to our claim that the PIP_3_/Ras patch controls the peripheral ring of SCAR by experimental manipulation of the signalling patch and of the peripheral ring of F‐actin. Disruption of the actin ring does not prevent signal patch formation. However, increasing patch size by disruption of a RasGAP leads to a concomitant increase in the peripheral ring of SCAR. These data substantiate our claim that the patch orchestrates the peripheral ring of SCAR.

*Reviewer #3:*

*[…] I would, however, suggest toning down some overstated conclusions that might be misleading for some readers.*

*1) At the end of the subsection “All PIP_3_ patches, whatever their origin, recruit SCAR to their periphery” the authors state that: "[…] any process that causes patches of PIP_3_ will cause recruitment of SCAR at the edges […]". This is a rather exaggerated conclusion based on colocalization data, especially when the authors later show that PIP_3_ is not needed for SCAR localization.*

This snippet and similar sentences that suggest that PIP_3_ recruits SCAR have been removed in the revised manuscript.

*2) At the end of the subsection “Macropinosomes in wild-type cells show the same as organisation as in axenic cells, but basal* PIP_3_*waves do not form” they also conclude "[…] that* PIP_3_*patches cause peripheral SCAR and actin rings to form […]". Again this is too strong a conclusion. No evidence is presented that* PIP_3_*causes the SCAR localization.*

The sentence has been removed in the revised manuscript where we have more carefully worded our conclusions.

[Editors' note: the author responses to the re-review follow.]

*[…] The manuscript has been improved but there are some remaining issues that need to be addressed before acceptance. No additional experimental work is needed, but the Discussion section should be revised to clarify points identified by reviewers 2 and 3.*

*1) "The authors should address what is known about the role of specific Racs that have been described as having a role in macropinocytosis, such as Rac1A and Rac1C and perhaps RacB, so as to remove some of the ambiguity from the presentation of the data and synthesis of the results. "*

Rac function has been primarily studied in *Dictyostelium* by overexpression of WT or constitutively active isoforms. Perturbation of Rac1 function, either by expressing constitutive active or dominant isoforms has a detrimental effect on macropinocytosis, suggesting that normal Rac1 function is essential for fluid phase endocytosis. Exactly how Rac1 is involved is unknown. Synthetic high levels of active Rac1 by overexpression of constitutive active isoforms greatly increases the number of circular ruffles, yet fluid phase uptake is decreased. Clearly, generation of ruffles alone is not sufficient of efficient uptake of fluid (Dumontier et al. 2000, Palmieri et al. 2000).

Conditional overexpression of RacC yields similar contrasting results; Phagocytosis is increased threefold in these cells, whereas fluid-phase uptake is reduced. These changes are mediated by PIP_3_, as pharmacological inhibition of PI3-Kinase by Wortmannin or LY294002 suppresses these effects (Seastone et al. 1998).

Conditional overexpression of RacB leads to numerical spherical surface protrusion, but cells also stop growing phagocytosis rates are reduced. It is thus not certain whether the protrusions represent unregulated attempts at macropinocytosis (Lee et al. 2003).

Overexpression of RacG also increases the rate of phagocytosis. Macropinocytosis was not measured. Similar as with RacG, the increased uptake is dependent on PIP_3_, as it can be inhibited using LY294002. Microscopically, the RacG isoform gives a hint at possible involvement in ring formation; RacG was found to be enriched at the rim of phagocytic cups (Somesh et al. 2006). No sensors for activated RacG currently exist, so it cannot be determined whether thus accumulated Rac is functional.

In summary, the results provide clues to involvement of Rac in macropinocytosis, but the used experiments are relatively crude and it is difficult to discriminate between specific effects and general disarray of cytoskeletal function caused by the mutations. As a result, there is no clear picture of how Rac is involved in macropinocytosis other than a general link with actin polymerization. A condensed report of these publications of known roles for Rac isoforms on macropinocytosis has been added to the Results section of the revised manuscript.

*2) Known roles for RasC or RasG in macropinocytosis, if any, should be mentioned.*

RasC and RasG are *Dictyostelium*-specific Ras isoforms. Ras in *Dictyostelium* has thus far mostly been researched in the context of gradient sensing and chemotaxis. However, several papers investigate the role of Ras isoforms on growth and fluid uptake. Relevant references have been added to the manuscript as follows:

“Two independent RasG- mutants are substantially impaired in growth in liquid medium, as previously described, but contrary to the earlier report (Khosla et al., 2000), both are also defective in fluid uptake. Compensation by other Ras proteins and genetic background differences may account for the discrepancy (Bloomfield et al., 2008; Bolourani et al., 2010). RasC null cells have no growth defect and a lesser defect in fluid uptake, while RasS (not tested here) may also contribute to macropinocytosis (Chubb et al., 2000).”

*3) Known roles for myosin 1 isoforms in macropinocytosis should be mentioned.*

We have expanded the discussion on myosin 1, adding 4 new references and including both genetic as microscopic evidence for a role of myosin-1 in macropinocytosis:

“Alternatively, SCAR might be moved to the periphery of Ras/PIP_3_ patches, perhaps by myosin-1 motors. […] In such a scenario, CARMIL could provide the link between myosin-1 and SCAR (Jung et al. 2001).”

*4) Citations of Hoeller et al. and Seastone et al. should be included with the reference to Clark et al. 2014.*

We assume that the referee means the first reference to Clark et al. 2014 where the effect of several genetic mutations on macropinocytosis are presented. Hoeller et al. 2013 studied the effects of several PI3K mutations, including fluid uptake. Seastone et al. 2001 first identified SCAR and immediately noticed an effect on macropinocytosis. Both papers are extremely relevant and were already cited elsewhere in the paper, but the referee is correct that they should also be included with the Clark et al. 2014 citation, where they fit perfectly. The Hoeller reference was included with the Clark reference. The Seastone reference was included at the end of the paragraph.

*5) Additional thoughts about the nature of the diffusion barrier that remains after latrunculin treatment are optional but encouraged.*

Sharply defined membrane domains continue to exist even after treatment with latrunculin, however, this does not imply that a diffusion gradient must exist. A combination of rapid synthesis of PIP_3_ on the patch and rapid degradation outside of the patch, likely supported by positive feedback loops, may also account for these domains. As such, speculation on the nature of the diffusion barrier that remains after latrunculin treatment may be pre-emptive. Nonetheless, we have somewhat expanded our discussion on diffusion barriers, including an extra reference providing evidence that membrane curvature can also restrict diffusion.